# Deciphering Attention Mechanisms: Optimization and Fenchel Dual Solutions

## Abstract

Attention has been widely adopted in many state-of-the-art deep learning models. While the significant performance improvements it brings have attracted great interest, the theoretical understanding of attention remains limited. This paper presents a new perspective on understanding attention by showing that it can be seen as a solver of a family of estimation problems. Specifically, we explore a convex optimization problem central to many estimation tasks prevalent in the development of deep learning architectures. Instead of solving this problem directly, we address its Fenchel dual and derive a closed-form approximation of the optimal solution. This approach results in a generalized attention framework, with the popular dot-product attention used in transformer networks being a special case. We show that T5 transformer has implicitly adopted the general form of the solution by demonstrating that this expression unifies the word mask and the positional encoding functions. Finally, we discuss how these new attention structures can be practically applied in model design and argue that the underlying convex optimization problem offers a principled justification for the architectural choices in attention mechanisms.

## 1 Introduction

Attention-based deep neural networks are now integrated into cutting-edge language models that have revolutionized a broad range of tasks: machine translation (Bahdanau et al., 2014; Luong et al., 2015), sentiment classification (Wang et al., 2016), image captioning (Xu et al., 2015) and unsupervised representation learning (Devlin et al., 2019), etc. Especially, attention plays a pivotal role in the construction of the transformer architecture (Vaswani et al., 2017), which has had a profound impact on the deep learning field.

Despite great empirical success, the design principle of attention has not been well studied in the literature, and there is no in-depth understanding of why attention-based models (e.g. BERT (Devlin et al., 2019)) have significantly better performance than other models. This gap in understanding limits practitioners' ability to effectively employ attention layers, posing challenges in developing new attention-based architectures.

In this paper, we offer a new perspective for understanding attention by showing that it is in fact a solver for a certain type of optimization problem that corresponds to an inference task. We give several examples, all of which can be characterized as follows: given 1) an unreliable estimate of the mean of an unknown distribution $p$ on $\mathbb{R}^d$ and 2) a preference distribution $u$ on $\mathbb{R}^d$ encoding beliefs on $p$'s selection, the inference task is to get a better estimate of $p$'s mean given its unreliable estimate and $u$. We derive a convex optimization problem that is abstracted from the task and solve it by instead solving its Fenchel dual (Rockafellar, 1970, p.104). Remarkably, the derived expression of the improved estimate of $p$ gives a generalized attention structure whose special case is equivalent to the popular dot-product attention (Luong et al., 2015) that is also applied in the transformer network (Vaswani et al., 2017). In addition, we show that our generalized attention expression has been implicitly adopted by T5 transformer (Raffel et al., 2020) as the expression unifies the concept of word masks and its positional encoding functions. Extra examples are given to show how the generalized attention structures can be used in practice, and a novel optimal transport (OT)-based attention is derived to show how our framework helps develop more general attention structures. Additionally, experiments are performed, which validate our theoretical work.

## 2 Related work

Since 2019, several authors have investigated the properties and working mechanisms of attention. This series of works mainly addresses whether the attention mechanism can serve as a proxy of saliency (Michel et al., 2019; Voita et al., 2019; Jain & Wallace, 2019; Wiegreffe & Pinter, 2019; Serrano & Smith, 2020; Vashishth et al., 2020). Most of these works obtain insights into the attention mechanism by performing empirical studies. The related methods include analyzing the behaviours of trained attention-based models (Clark et al., 2019), pruning a few heads, analyzing the effects of altering the attention weights (Michel et al., 2019; Voita et al., 2019), or a mixture of these (Jain & Wallace, 2019; Vashishth et al., 2020).

Beyond empirical understanding, theoretical results by Brunner et al. (2019) and Hahn (2020) indicate that self-attention layers are not identifiable, meaning multiple combinations of attention weights can yield equally good predictions. This non-uniqueness complicates interpretability. Additionally, Tsai et al. (2019) reformulated attention using kernel theory, showing it can be viewed as applying a kernel smoother over the inputs. Recent studies have also explored the expressivity of attention (Dong et al., 2021; Baldi & Vershynin, 2022; Mahdavi et al., 2024). To understand the underpinning inductive bias of attention, Sahiner et al. (2022) have investigated convex-relaxations through the lens of convex duality by replacing the softmax function with element-wise nonlinear functions. While our work views the problem through a similar lens, the framework covers the unaltered attention architecture and focuses more on the design motivation of attention and its generalization.

Another important approach to understanding attention is to analyze its asymptotic behaviour when the number of heads and the network width approach infinity (Yang, 2019; Hron et al., 2020). In this limit, the entire network behaves as a Gaussian process (Lee et al., 2018) allowing for closed-form characterizations not available in the finite regime. Since 2021, several theoretical works have explored attention outside this asymptotic regime. Lu et al. (2021) set up a simple attention-based classification model and derive a closed-form relationship between the word's embedding norm and the product of its key and the query. They empirically show that such a relationship also exists in a more practical configuration. Similarly, Jelassi et al. (2022); Li et al. (2023); Deora et al. (2024) characterize optimization and generalization properties for gradient-descent training. Ramsauer et al. (2021) established an equivalence between attention and a newly proposed Hopfield network with continuous states, demonstrating that the new Hopfield network's update rule is equivalent to the attention mechanism used in transformers (Vaswani et al., 2017).

## 3 Setup of a design problem

We consiser a prediction task: given an input $\mathbf{X}$, predict an output quantity $\mathbf{Y} = (Y^{(1)}, Y^{(2)}, \ldots, Y^{(K)})$, including $K$ components. We will present several machine-learning problems and show they can be unified and abstracted into a mean estimation problem. Specifically, the goal is to estimate the mean of a distribution $p$, given a prototype of $p$ and a noisy estimate of the discrepancy between their means. By framing the problem in this way, we can devise a unified convex optimization framework to address these various scenarios. The solutions derived under this framework yield attention-like structures, which can be used to tackle the original prediction tasks. Furthermore, plugging in various functions for closeness constraints, we recover the original dot-product attention (Sec 6) and derive a variant with added properties (Sec 9).

**Translation Problem (TP).** In this problem, the input $\mathbf{X}$ is a sentence, or a sequence of words, in the source language. Output $\mathbf{Y}$ is the sequence of words in the target sentence, where $Y^{(k)}$ is the $k^{\text{th}}$ word.

**Image Captioning (IC).** In this problem, the input $\mathbf{X}$ is a raw image and output $\mathbf{Y}$ is the sequence of words in the caption, where $Y^{(k)}$ is the $k^{\text{th}}$ word.

**Filling in the Blanks Task (FB)**. This task has been used to train the BERT model (Devlin et al., 2019). The input $\mathbf{X}$ is a sequence of words with a certain percentage of words masked. The output $\mathbf{Y}$ are the predicted masked words, where $Y^{(k)}$ denotes the $k^{\text{th}}$ masked one.

The objective of any of these problems and that we address in this paper is to learn a function $\mathcal{F}$, mapping from the space of $\mathbf{X}$ to the space of $\mathbf{Y}$ so that $\mathbf{Y} = \mathcal{F}(\mathbf{X})$. We will denote by $F^{(k)}$ the part of $\mathcal{F}$ responsible

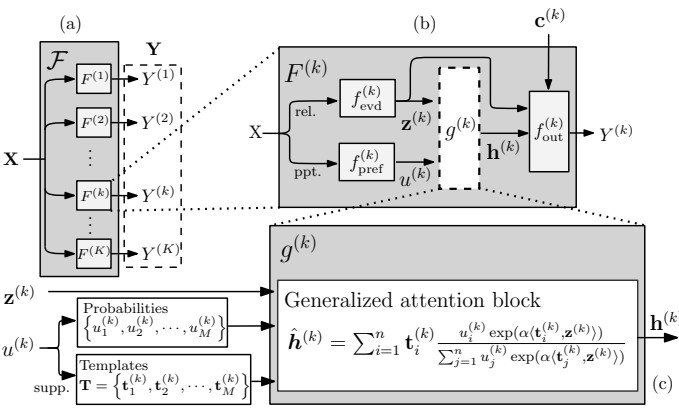

Figure 1: A conceptual graph of the deep learning model that we work with. The block $g^{(k)}$ is the one we will investigate. (a) shows the general structure of a sequence generation model, with $F^{(k)}$ responsible for the $k$-th output. Our focus is on the architecture in (b), where $F^{(k)}$ contains a component $g^{(k)}$ that infers a distribution's mean $\mathbf{h}^{(k)}$ based on its noisy estimations from two aspects: its preference (prior) distribution $u^{(k)}$ and a noisy estimation of its mean shift $\mathbf{z}^{(k)}$ from $u^{(k)}$'s. We show that $g^{(k)}$ should implement the expression in (c), which includes the dot-product attention as a special case (Luong et al., 2015).

for predicting $Y^{(k)}$ (Fig 1a), namely, $Y^{(k)} = F^{(k)}(X)$. Although we here express $\mathcal{F}$ as separate functions $(F^{(1)}, F^{(2)}, \ldots, F^{(K)})$, we note that it is in fact possible that different $F^{(k)}$'s share some component in common. Without loss of generality, we now focus on the design of $F^{(k)}$.

## 3.1 The Design Problem

In deep learning research, a typical approach to solving the three running tasks is first to use a neural network to extract vector representations $\{\mathbf{t}_1^{(k)}, \mathbf{t}_2^{(k)}, \ldots, \mathbf{t}_M^{(k)}\} \subseteq \mathbb{R}^d$ of $\mathbf{X}$, which are referred as templates. Collectively, we will denote the set $\{\mathbf{t}_1^{(k)}, \mathbf{t}_2^{(k)}, \ldots, \mathbf{t}_M^{(k)}\}$ of templates by $\mathbf{T}^{(k)}$.[1] (If $\mathbf{X}$ are words, typical choices of neural network include RNN, LSTM, etc. If $\mathbf{X}$ is an image, a typical choice is CNN.) Let $\mathcal{A} \subseteq \mathbb{R}^d$ denote the space containing all templates. For each $Y^{(k)}$, some mechanism $g^{(k)}$ is needed to adaptively combine the representations of $\mathbf{X}$ to obtain $\mathbf{h}^{(k)}$, which is then fed into a classifier $f_{\text{out}}^{(k)}$ to predict $\mathbf{Y}^{(k)}$.

To obtain an idea of how to produce $\mathbf{h}^{(k)}$, consider **TP** task, where $\mathbf{h}^{(k)}$ corresponds to a vector (also known as embedding) representing the $k$-th word in the target sentence, and $\mathbf{T}^{(k)} = \{\mathbf{t}_1^{(k)}, \mathbf{t}_2^{(k)}, \ldots, \mathbf{t}_M^{(k)}\}$ are the ones of the source sentence.[2] Then, the inference of $\mathbf{h}^{(k)}$ corresponds to combining the semantic meanings encoded in $\mathbf{T}^{(k)}$ to produce the $k$-th word embedding in the target sentence. This can be simply modelled as

$$\boldsymbol{h}^{(k)} = \int_\Omega \mathbf{t} \, p^{(k)}(\mathbf{t}) \, \mathrm{d}\mathbf{t} \quad \text{s.t. } p^{(k)}(\mathbf{t}) \geq 0 \text{ for all } \mathbf{t} \in \mathbf{T}^{(k)} \text{ and } \int_\Omega p^{(k)}(\mathbf{t}) \, \mathrm{d}\mathbf{t} = 1, \quad (1)$$

where $\Omega = \mathbf{T}^{(k)}$. That is, $\mathbf{h}^{(k)}$ is a convex combination of $\mathbf{t} \in \mathbf{T}^{(k)}$, or equivalently, the mean of an unknown distribution $p^{(k)}$ on $\mathbf{T}^{(k)}$. For generality, our following discussion extends the support of $p^{(k)}$ to all possible templates by setting $\Omega = \mathcal{A}$. In Sec 9, we show that this extension enables optimal transport-based attention, taking into account words having similar embeddings in $\mathbf{T}^{(k)}$. That is, even if a word is not present in the source sentence, its embedding will still be optimized if it has a similar embedding in $\mathbf{T}^{(k)}$.

In practice, the cardinality of $\mathcal{A}$ may be huge or infinite; therefore, it is important to design a mechanism that allows the users to inject prior knowledge to guide the production of $\mathbf{h}^{(k)}$. For example, in **TP** task, $\mathcal{A}$ would be the set of all word embeddings, which could contain more than 10K elements. However, $\mathbf{h}^{(k)}$ should largely depend on the templates associated with the words (in the input sentence) having similar locations to the $k$-th word in the target sentence. If we could effectively inject this prior information, the inference task would be largely simplified. One natural way to do so is to use a neural network module $f_{\text{pref}}^{(k)}$ to propose a prototype of $p^{(k)}$, referred to as the preference distribution $u^{(k)}$, and let $p^{(k)}$ be close $u^{(k)}$. Specifically $u^{(k)}$ puts non-zero probability masses on templates $\mathbf{t}_1^{(k)}, \mathbf{t}_2^{(k)}, \ldots, \mathbf{t}_M^{(k)}$, and their probabilities are respectively $u_1^{(k)}, u_2^{(k)}, \ldots, u_M^{(k)}$ (which sum to 1). For **TP**, $u^{(k)}$ is expected to have larger values for the words in a similar

---

[1]We add the superscript $k$ to note that the inference of $Y^{(k)}$ does not necessarily share the set of templates.
[2]We mainly use **TP** to motivate the design and discussion. In Sec 3.2, we show the same ideas also apply to **IC** and **FB**.

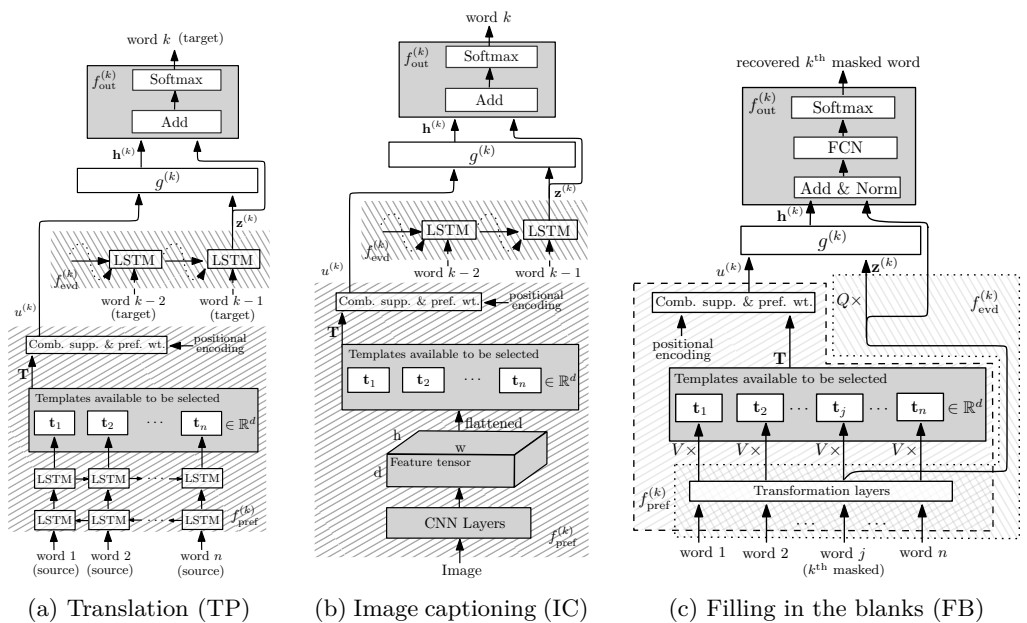

Figure 2: The model architectures of the three running examples. For the $f_{\text{evd}}^{(k)}$ in (a) and (b), the dashed links exist throughout the training and are replaced by the dotted ones in the generation stage.

location of the $k$-th word of the target sentence. The preference distribution $u^{(k)}$ is considered as a good approximation of $p^{(k)}$, in the sense that the support of $p^{(k)}$ is contained in the set $\mathbf{T}^{(k)}$ of templates. Note that if $\mathbb{R}^d$ is the word embedding space for a large vocabulary, and if the size $M$ of the template set $\mathbf{T}^{(k)}$ is relatively small, then restricting the support of $p^{(k)}$ to be within $\mathbf{T}^{(k)}$ imposes a strong constraint on $p^{(k)}$. On the other hand, $u^{(k)}$ is not a sufficiently accurate approximation of $p^{(k)}$, in the sense that $u^{(k)}$ may assign probabilities to $\mathbf{T}^{(k)}$ somewhat differently. For example, in **TP**, the choice of $Y^{(k)}$ depends on both $\mathbf{X}$ and the already generated words $Y^{(i<k)}$.[3] While $u^{(k)}$ provides a strong prior that $p^{(k)}$ should mainly focus on the words appearing in the source sentence, it is inherently tough for $u^{(k)}$ to capture the semantic evolution in $Y^{(i<k)}$. The difficulty shifts the mean $\boldsymbol{\mu}^{(k)}$ of $u^{(k)}$ from the mean $\mathbf{h}^{(k)}$ of $p^{(k)}$.

To alleviate the problem, we need another piece of information $\mathbf{z}^{(k)} \in \mathbb{R}^d$ that is generated by another network module $f_{\text{evd}}^{(k)}$ and provides information regarding the mean shift. In **TP**, $\mathbf{z}^{(k)}$ depends on $Y^{(i<k)}$.) In particular, we assume that $\mathbf{z}^{(k)}$ is a noisy version of the shift, more precisely,

$$\mathbf{z}^{(k)} = \mathbf{h}^{(k)} - \boldsymbol{\mu}^{(k)} + \epsilon, \tag{2}$$

where $\epsilon \sim \mathcal{N}(\mathbf{0}, \sigma^2 \mathbf{I})$ is a spherical Gaussian noise in $\mathbb{R}^d$ with covariance $\sigma^2 \mathbf{I}$. We refer to $\mathbf{z}^{(k)}$ as the evidence.

We summarize the problem setup in Fig 1b. Then the design problem is *to construct a function, or a network block, $g$, which infers the unknown distribution $p^{(k)}$ and hence its mean $\mathbf{h}^{(k)}$ based on the evidence $\mathbf{z}^{(k)}$ and the preference distribution $u^{(k)}$.*

### 3.2   Additional examples

Having demonstrated how the setup applies to the translation problem (**TP**), we will now illustrate its applicability to the other two examples.

**Image Captioning (IC).** The caption generation model in Fig 2b has a similar architecture adopted by Xu et al. (2015), where $f_{\text{pref}}^{(k)}$ extracts templates from the image using a CNN. In this task, a word's position is independent of the object's location, so all CNN-extracted templates have the same preference weight.

---

[3]We assume the sentence generation process is Markovian. More details are given in Sec 3.2.

Similar objects in the image have similar CNN features. Allowing non-$\mathbf{T}$ templates to influence $\mathbf{h}^{(k)}$ could introduce irrelevant information, harming word inference accuracy. To improve $\mathbf{h}^{(k)}$ estimation, we constrain $p^{(k)}$ to have support only within $u^{(k)}$. As generation progresses, $\mathbf{h}^{(k)}$ should evolve to provide relevant image information for the next word. This semantic evolution is captured by $\mathbf{z}^{(k)} = f_{\text{evd}}^{(k)}$, which predicts the shift of $\boldsymbol{\mu}^{(k)}$ from $\mathbf{h}^{(k)}$. So $\boldsymbol{\mu}^{(k)} + \mathbf{z}^{(k)}$ estimates $\mathbf{h}^{(k)}$ and should be close to it, as should $u^{(k)}$ and $p^{(k)}$.

**Filling in the Blanks Task (FB)**. For filling-in-the-blank tasks, consider a BERT-like model (Fig 2c) where $f_{\text{pref}}^{(k)}$ and $f_{\text{evd}}^{(k)}$ share transformation layers common to NLP tasks. $f_{\text{pref}}^{(k)}$ applies a linear map $V$ to the output sequence of the previous layer to form the template set $\mathbf{T}$ supporting $u^{(k)}$, with preference weights specified by positional encoding. Concurrently, $\mathbf{z}^{(k)} = f_{\text{evd}}^{(k)}$ estimates the shift of $\mathbf{h}^{(k)}$ from the mean $\boldsymbol{\mu}^{(k)}$ due to local variation. As before, we need $\boldsymbol{\mu}^{(k)} + \mathbf{z}^{(k)}$ close to $\mathbf{h}^{(k)}$ and $p^{(k)}$ close to $u^{(k)}$. Notably, the formulation of the problem is based on the assumption that the network modules $f_{\text{evd}}^{(k)}$ and $f_{\text{pref}}^{(k)}$ are fixed and generate $\mathbf{z}^{(k)}$ and $u^{(k)}$ satisfying the above-assumed properties. In reality, $f_{\text{evd}}^{(k)}$ and $f_{\text{pref}}^{(k)}$ are obtained via training. However, we argue that if $g$ is made to satisfy our design objective, we can at least *interpret* $f_{\text{evd}}^{(k)}$ and $f_{\text{pref}}^{(k)}$ obtained from training as serving to produce $\mathbf{z}^{(k)}$ and $u^{(k)}$ with our desired properties.

## 4 Formulation of an optimization problem

The discussion made in the previous section implies that the key optimization problem we are about to focus on should ensure

1. $\mathbf{h}^{(k)}$ is not too far from $\boldsymbol{\mu}^{(k)} + \mathbf{z}^{(k)}$, where $\mathbf{h}^{(k)}$ is constructed by $p^{(k)}$ according to (1) and $\boldsymbol{\mu}^{(k)}$ is the mean of the preference distribution $u^{(k)}$.

2. $p^{(k)}$ is close to $u^{(k)}$ while $p^{(k)}$'s support is a subset of $u^{(k)}$'s.

These two desiderata prompt us to optimize:

$$\min_{p} \frac{\alpha}{2} \left\| (\boldsymbol{\mu} + \mathbf{z}) - \int_{\mathbb{R}^d} \mathbf{a} p(\mathbf{a}) \, d\mathbf{a} \right\|^2 + \mathcal{K}(p, u) \tag{3}$$

where $\alpha > 0$ is responsible for the relative strength of the two terms (and can be interpreted as the reliability of $\boldsymbol{\mu} + \mathbf{z}$), $\mathcal{K}(p, u)$ denotes the KL divergence of $u$ from $p$.[4] By definition, $\mathcal{K}(p, u)$ has a finite value if and only if $p$ has zero values outside the support of $u$. Thus, both requirements in the second desideratum are satisfied by using the KL divergence as a measure for the closeness of $p$ and $u$. Let $\tilde{p}$ be the minimizer of (3). The estimate of $\mathbf{h}$ is

$$\hat{\mathbf{h}} = \int_{\mathbb{R}^d} \mathbf{a} \tilde{p}(\mathbf{a}) \, d\mathbf{a}. \tag{4}$$

Naturally, this optimization problem can be derived from three different, though related perspectives. Below, we present a less commonly known view that demonstrates how $\alpha$ affects the optimal solution from a hard constraint perspective. The maximum likelihood and Bayesian perspectives are included in Appx B.

**A Maximum Entropy on the Mean Perspective.** Consider a problem that seeks a distribution $p$ such that the expectation $\int_{\mathbb{R}^d} \mathbf{a} p(\mathbf{a}) \, d\mathbf{a}$ is not far from $\boldsymbol{\mu} + \mathbf{z}$. Namely, we require $\left\| (\boldsymbol{\mu} + \mathbf{z}) - \int_{\mathbb{R}^d} \mathbf{a} p(\mathbf{a}) \, d\mathbf{a} \right\|^2 \leq \frac{1}{2\alpha}$. Given $\mathbf{z}$, there are infinitely many $p$'s that satisfy the constraints, making it difficult to select the "best" $p$. A technique in information theory, maximum entropy on the mean (MEM) (Rioux et al., 2020; Gamboa, 1989), addresses this by selecting the best guess of the ground truth $p^*$ that satisfies the constraint and minimizes the KL divergence:

$$\tilde{p} = \operatorname*{argmin}_{p} \mathcal{K}(p, u) \quad \text{s.t.} \quad \left\| (\boldsymbol{\mu} + \mathbf{z}) - \int_{\mathbb{R}^d} \mathbf{a} p(\mathbf{a}) \, d\mathbf{a} \right\|^2 \leq \frac{1}{2\alpha},$$

which also minimizes (3) according to (Rioux et al., 2020, Eq (18)) and (Borwein & Lewis, 1992, Cor 4.9).

---

[4]As we will focus on a single step of sequence predictions, we simplify our notations by omitting superscript $(k)$ in the rest of our discussions.

## 5 A motivating example to find the optimal solution

To better illustrate our method for solving (3), we first examine a special case where the preference distribution $u$ follows a spherical Gaussian distribution, specifically $u \sim \mathcal{N}(\boldsymbol{\mu}, I_d)$. In this scenario, the convex problem can be solved in closed form. The derivation provides valuable insights into how the problem can be approached in a general context, as we will demonstrate in Sec 6.

Let $\mathbf{b} = \boldsymbol{\mu} + \mathbf{z}$ serve as an unreliable observation of $\mathbf{h}_p$. Rioux et al. (Rioux et al., 2020) prove, via Fenchel duality (Rockafellar, 1970, p.104) that the minimizer $p^*$ of (3) takes the form

$$p^*(\mathbf{a}) = \frac{u(\mathbf{a}) \exp\langle \mathbf{a}, \boldsymbol{\lambda}^* \rangle}{\int u(\mathbf{a}') \exp\langle \mathbf{a}', \boldsymbol{\lambda}^* \rangle \, \mathrm{d}\mathbf{a}'}, \tag{5}$$

where

$$\boldsymbol{\lambda}^* = \operatorname*{argmax}_{\boldsymbol{\lambda} \in \mathbb{R}^d} \langle \mathbf{b}, \boldsymbol{\lambda} \rangle - \frac{1}{2\alpha} \|\boldsymbol{\lambda}\|^2 - \log \int u(\mathbf{a}) \exp\langle \mathbf{a}, \boldsymbol{\lambda} \rangle \, \mathrm{d}\mathbf{a}. \tag{6}$$

Note that $\int u(\mathbf{a}) \exp\langle \mathbf{a}, \boldsymbol{\lambda} \rangle \, \mathrm{d}\mathbf{a} = \exp(\langle \boldsymbol{\mu}, \boldsymbol{\lambda} \rangle + \frac{1}{2} \|\boldsymbol{\lambda}\|^2)$ as it is the moment generating function (MGF) of $u \sim \mathcal{N}(\boldsymbol{\mu}, I_d)$. Substituting the expression into (6) followed by setting the derivative with respect to $\boldsymbol{\lambda}$ to zero yields $\boldsymbol{\lambda}^* = \frac{\alpha}{\alpha+1}(\boldsymbol{b} - \boldsymbol{\mu})$. By (5), $p^*(\mathbf{a}) \propto \exp(-\frac{1}{2} \|\mathbf{a} - \boldsymbol{\mu}\|^2 + \langle \mathbf{a}, \boldsymbol{\lambda}^* \rangle) \propto \exp(-\frac{1}{2} \|\mathbf{a} - (\boldsymbol{\mu} + \boldsymbol{\lambda}^*)\|^2)$. Substituting $\boldsymbol{\lambda}^* = \frac{\alpha}{\alpha+1}(\boldsymbol{b} - \boldsymbol{\mu})$ into it implies that $p^*$ follows a Gaussian distribution $\mathcal{N}(\frac{1}{1+\alpha}\boldsymbol{\mu} + \frac{\alpha}{1+\alpha}\boldsymbol{b}, I_d)$. Thus, our estimate of $\mathbf{h}_p$ is $\frac{1}{1+\alpha}\boldsymbol{\mu} + \frac{\alpha}{1+\alpha}\mathbf{b}$.

The value $\alpha$ in (3) can also be considered as a measure of the reliability of the noisy observation $\boldsymbol{b}$, where a smaller $\alpha$ implies a less reliable $\boldsymbol{b}$. Then, the estimate of $\boldsymbol{h}_p$ should be less affected by $\boldsymbol{b}$ as $\alpha$ approaches zero, which is well captured by our derived expression $\frac{1}{1+\alpha}\boldsymbol{\mu} + \frac{\alpha}{1+\alpha}\mathbf{b}$. We will also see this relationship in a more general setting in our subsequent discussions. While a more complicated analysis is involved, the underlying principles are essentially the same.

In Sec 6, we focus on a similar optimization problem that estimates $\mathbf{h}_p$ assuming that $u$ is instead a discrete distribution. By solving the optimization problem, we derive a closed-form approximation for the estimate of $\mathbf{h}_p$, via Fenchel duality. The approximation then gives a generalized attention layer structure as shown in Fig 1. A special case of it is equivalent to the familiar dot-product attention (Luong et al., 2015) that is also adopted in transformers (Vaswani et al., 2017). Moreover, we will show that T5 transformer (Raffel et al., 2020) implicitly adopts our generalized attention expression.

## 6 Attention as inference via Fenchel duality

Now we present how to solve (3) with general $u$, where the solution yields the standard attention mechanism. Rioux et al. proved that the optimization problem stated in (3) has the following Fenchel dual:

**Theorem 1.** *The dual of (3) is given by*

$$\max_{\lambda \in \mathbb{R}^d} \left\{ \langle \boldsymbol{\lambda}, \boldsymbol{\mu} + \mathbf{z} \rangle - \frac{1}{2\alpha} \|\boldsymbol{\lambda}\|^2 - \log M(\boldsymbol{\lambda}) \right\}, \tag{7}$$

*where*

$$M(\boldsymbol{\lambda}) = \int_{\mathbb{R}^d} u(\mathbf{a}) \exp\langle \mathbf{a}, \boldsymbol{\lambda} \rangle \, \mathrm{d}\mathbf{a}. \tag{8}$$

*Given a maximizer $\boldsymbol{\lambda}^*$ of (7), one can recover the minimizer $\tilde{p}$ of (3) via*

$$\tilde{p}(\mathbf{a}) = \frac{u(\mathbf{a}) \exp\langle \mathbf{a}, \boldsymbol{\lambda}^* \rangle}{\int_{\mathbb{R}^d} u(\mathbf{a}') \exp\langle \mathbf{a}', \boldsymbol{\lambda}^* \rangle \, \mathrm{d}\mathbf{a}'}. \tag{9}$$

By Theorem 1, the estimated $\mathbf{h}$ defined in (4) can be re-written as

$$\hat{\mathbf{h}} = \int_{\mathbb{R}^d} \mathbf{a}\tilde{p}(\mathbf{a}) \, \mathrm{d}\mathbf{a} = \int_{\mathbb{R}^d} \mathbf{a} \frac{u(\mathbf{a}) \exp\langle \mathbf{a}, \boldsymbol{\lambda}^* \rangle}{\int_{\mathbb{R}^d} u(\mathbf{a}') \exp\langle \mathbf{a}', \boldsymbol{\lambda}^* \rangle \, \mathrm{d}\mathbf{a}'} \, \mathrm{d}\mathbf{a}, \tag{10}$$

where $\boldsymbol{\lambda}^*$ is a maximizer of (7).

In general, $\boldsymbol{\lambda}^*$ does not have a closed-form expression in terms of $\alpha$, $u$ and $\mathbf{z}$, and a standard paradigm is to search for it using gradient ascent-based methods. In this paper, we will not search for $\boldsymbol{\lambda}^*$ in this way; instead, we will derive a closed-form expression to approximate it. Remarkably, this takes the form of the generalized attention presented in Fig 1. Note that $M(\boldsymbol{\lambda})$ in (8) equals $\mathbb{E}_u[\exp\langle W, \boldsymbol{\lambda} \rangle]$, the expectation of the random variable $\exp\langle W, \boldsymbol{\lambda} \rangle$ where $W$ has the probability distribution $u$. The expectation is just the moment generating function (MGF) of $W$, and the value $\log M(\boldsymbol{\lambda})$ is called the cumulant of $W$ (McCullagh, 1987, p.26), which has an expansion (McCullagh, 1987, (2.4))

$$\log M(\boldsymbol{\lambda}) = \langle \boldsymbol{\mu}, \boldsymbol{\lambda} \rangle + \frac{1}{2} \langle \boldsymbol{\lambda}, \Sigma\boldsymbol{\lambda} \rangle + o(\|\boldsymbol{\lambda}\|^2), \tag{11}$$

with $\boldsymbol{\mu} = \int \mathbf{a}u(\mathbf{a}) \, \mathrm{d}\mathbf{a}$ and $\Sigma = \int (\mathbf{a} - \boldsymbol{\mu})(\mathbf{a} - \boldsymbol{\mu})^T u(\mathbf{a}) \mathrm{d}\mathbf{a}$ respectively denote the expectation and the variance-covariance matrix of $W$. Note that the expansion implicitly assumes that random variable $W$ following distribution $u$ has bounded moments. (Derivation of (11) is given in Appx A.)

Now we assume that $\alpha$ is small and we argue that this assumption is justified in practice. For instance, in the translation task, all words in the dictionary can serve as candidate templates, which could be more than 10,000, but $u$ reduces this size to the length of the source sentence (usually less than tens of words). The inference of $p$ should strongly anchor around this prior information; consequently the information provided by $\mathbf{z}$ should weigh less. On the other hand, $\mathbf{z}$ can hardly provide an accurate estimate of the mean shift, since the generation of $\mathbf{z}$ is often ignorant of the templates selected by $u$ (for example, in the example translation and image captioning models) or generated by a low-capacity module (as in the example filling-in-the-blank model). For these reasons, one should de-emphasize the constraint imposed by $\mathbf{z}$ and thus choose a small $\alpha$.

When $\alpha$ is picked to be small enough (see (7)), the optimization of $\boldsymbol{\lambda}$ gets a large penalty on its L2 norm and thus, $\|\boldsymbol{\lambda}^*\|$ is close to zero. Then, by (11), we have

$$\log M(\boldsymbol{\lambda}^*) \approx \langle \mu, \boldsymbol{\lambda}^* \rangle + \frac{1}{2} \langle \boldsymbol{\lambda}^*, \Sigma\boldsymbol{\lambda}^* \rangle. \tag{12}$$

Note that the approximation becomes exact for any $\alpha > 0$ if $u$ is Gaussian, which is the case of the motivating example in Sec 5. Substituting (12) into (7) followed by setting the derivative with respect to $\boldsymbol{\lambda}$ to zero yields

$$\boldsymbol{\lambda}^* = \alpha(I_d + \alpha\Sigma)^{-1}\mathbf{z}, \tag{13}$$

where $I_d$ denotes the $d \times d$ identity matrix.[5] As $\alpha$ is assumed close to zero, (13) is further reduced to

$$\boldsymbol{\lambda}^* = \alpha\mathbf{z}. \tag{14}$$

Plugging the expression into (10) gives the result stated as follows:

**Theorem 2.** *Given $u$ with bounded moments, for a small enough $\alpha > 0$, the estimated $\mathbf{h}$ defined in (4) can be approximated by*

$$\hat{\mathbf{h}} = \int_{\mathbb{R}^d} \mathbf{a} \frac{u(\mathbf{a}) \exp(\alpha\langle \mathbf{a}, \mathbf{z} \rangle)}{\int_{\mathbb{R}^d} u(\mathbf{a}') \exp(\alpha\langle \mathbf{a}', \mathbf{z} \rangle) \, \mathrm{d}\mathbf{a}'} \, \mathrm{d}\mathbf{a}. \tag{15}$$

For the case that $u$ is a discrete distribution with support $\{\mathbf{t}_1, \mathbf{t}_2, \ldots, \mathbf{t}_n\}$ and the preference probability $\{u_1, u_2, \ldots, u_n\}$, (15) becomes simply

$$\hat{\mathbf{h}} = \sum_{i=1}^n \mathbf{t}_i \frac{u_i \exp(\alpha\langle \mathbf{t}_i, \mathbf{z} \rangle)}{\sum_{j=1}^n u_j \exp(\alpha\langle \mathbf{t}_j, \mathbf{z} \rangle)}. \tag{16}$$

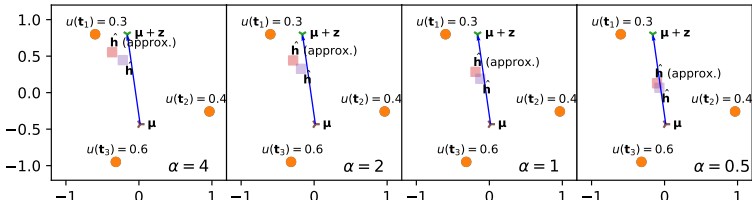

Figure 3: The approximation of $\hat{\mathbf{h}}$ for different choices of $\alpha$. The dots in orange compose the support of discrete $u$ with the preference weights labelled above. The dark blue arrow starting from the mean $\mu$ of $u$ denotes the evidence $\mathbf{z}$. The red square marks the $\hat{\mathbf{h}}$ constructed by (10) with the $\boldsymbol{\lambda}^*$ maximizing (7), while the purple one marks the $\hat{\mathbf{h}}$ approximated by (16). As we can observe, (16) gives a precise approximation of $\hat{\mathbf{h}}$ when $\alpha$ is sufficiently small.

In Fig 3, we set $d = 2$ and visualize the approximation of $\mathbf{h}$ for various selections of $\alpha$. We observe that, as $\alpha$ decreases, (16) outputs a better approximation of $\hat{\mathbf{h}}$. Besides, as a decreasing $\alpha$ implies a less reliable $\boldsymbol{\mu} + \boldsymbol{z}$, $\boldsymbol{h}$ is less affected by $\boldsymbol{\mu} + \boldsymbol{z}$ and gets close to $\boldsymbol{\mu}$. Note that our results do not suggest that $\alpha$ should be arbitrarily close to zero for a perfect approximation (which leaves $\mathbf{z}$ useless). Fig 3 shows a good approximation is achieved when $\alpha = 0.5, 1$. And for these two choices, $\hat{\boldsymbol{h}}$ still significantly deviates from $\boldsymbol{\mu}$ (corresponding to the case when $\alpha = 0$ and $\mathbf{z}$ is useless). Thus, $\mathbf{z}$ still largely affects the final estimation results.

The derived solution in (16) aligns with the original attention mechanisms discussed by Bahdanau et al. (2014) and Luong et al. (2015), where $u$ is set to a uniform distribution. These models have incorporated most of the crucial components of the modern transformer architecture. In Sec 8, we will demonstrate that (16) also extends to more contemporary architectures, such as the BERT model (Devlin et al., 2019) and T5 (Raffel et al., 2020). Furthermore, we will show that a good approximation can be achieved in practice by comparing the accurate solution with its approximated counterpart used in these pretrained models.

## 7 Discussion

In Section 6, we derived an alternative expression of $\hat{\mathbf{h}}$ defined in (4) by solving the Fenchel dual of the optimization problem (3). Although the expression is not in closed form, as we are only interested in the case when $\alpha$ is small, a closed-form approximation of $\hat{\mathbf{h}}$ is derived in Theorem 2 and reduced to the form stated in (16) when considering a discrete distribution $u$.

As we pointed out, the block $g$ in Fig 2a, Fig 2b and Fig 2c is expected to find the inferred $\tilde{p}$ minimizing (3) followed by plugging it into (4) to construct $\hat{\mathbf{h}}$. Thus, one can complete the architecture designs of the three running examples by replacing $g$ with a network layer implementing (16), namely, the structure in Fig 1c.

**The relationship between the optimal solution and attention models.** Remarkably, the expression stated in (16) gives a generalized attention block. In particular, based on our framework, researchers can customize the implementations of $f_{\text{evd}}^{(k)}$ and $f_{\text{pref}}^{(k)}$ to generate $\mathbf{z}$ and $u$ and feed them into (16) to get an attention-like network architecture.[6]

For instance, by setting $u_i = \frac{1}{n}$ for all $i$, the expression is equivalent to the well known dot-product attention (Luong et al., 2015), which is also applied in the transformer network (Vaswani et al., 2017). The equivalence of the expression of $\hat{\mathbf{h}}$ and the dot-product attention layer tells us: (a) *by applying a dot-product attention layer in a model, we essentially ask the model to perform an optimization task defined in (3) and construct the output according to (4).* (b) *the derivation of h depends on two relatively independent pieces of information: a preference distribution given the global information and an estimate of the output's deviation from the preference distribution's mean according to some local information. This suggests that the design of attention-based model can be decomposed into two parts that respectively estimate these two values.*

---

[5]When $\Sigma = I_d$, (13) becomes $\boldsymbol{\lambda}^* = \alpha(I_d + \alpha I_d)^{-1}\mathbf{z} = \frac{\alpha}{1+\alpha}\mathbf{z}$. By (2), $\mathbf{b} = \mathbf{h} + \epsilon = \mathbf{z} + \boldsymbol{\mu}$. Thus, $\boldsymbol{\lambda}^* = \frac{\alpha}{1+\alpha}(\boldsymbol{b} - \boldsymbol{\mu})$ recovers the expression of $\boldsymbol{\lambda}^*$ in the motivating example.

[6]Potential selectionss of $f_{\text{evd}}^{(k)}$ and $f_{\text{pref}}^{(k)}$ includes constant functions, fixed formulas and neural networks.

**The model consisting of a stack of attention layers.** Although our discussion focuses on the case that contains a single attention layer, any attention layer $\mathcal{L}$ in an attention stack fits our framework (see Fig 1). In particular, all the attention layers closer to the input $X$ than $\mathcal{L}$ can be grouped into the functions $f_{\text{pref}}$ or $f_{\text{evd}}$. For those layers that take the current layer's output as input, we can group them into $f_{\text{out}}$, where $\mathbf{c}$ may contain the outputs of other attention layers working in parallel.

**Multi-head attention.** For clarity, our derivation does not account for multi-head attention scenarios. In essence, an $n$-head attention structure can be viewed as having $n$ distinct estimations of mean shift estimates. Consequently, the outputs of $n$-head attention can be interpreted as the solutions to $n$ underlying convex problems, which are subsequently stacked together at the end of the inference processes.

**T5 transformer implicitly adopts the generalized attention structure.** Recent studies in NLP have shown that T5 (Raffel et al., 2020) can achieve state-of-the-art performance for many NLP benchmarks, including text summarization, classification, question answering, etc. While their transformer implementations are quite similar to the original transformer architecture (Vaswani et al., 2017; Devlin et al., 2019), they adopt trainable relative position embeddings to replace the sinusoidal position signals.[7] The modification provides the model with extra flexibility to encode the positional information with little computational cost.

We will see that in comparison to the original transformer implementation, T5 transformer can be seen as a natural realization of the generalized attention in (16), where the preference weights $u$ unifies the concepts of word masks and T5's positional encoding functions. Thus, the usefulness and the validity of our framework are well-supported by the state-of-the-art performance of T5 in many NLP tasks (Raffel et al., 2020).

Consider the running example: filing in the blanks, with the preference distribution

$$u(\mathbf{t}_i) = \begin{cases} 0 & \text{if the } i^{\text{th}} \text{ word is masked} \\ \exp(b_{j-i})/Z & \text{otherwise,} \end{cases} \tag{17}$$

where $Z$ is a normalizing constant and $b_{j-i}$ is a trainable scalar that only depends on the relative position of word $i$ and word $j$ (which is the $k^{\text{th}}$ masked word that we are inferring). Substituting such $u$ into (16) with $\alpha = 1$ yields

$$\hat{\mathbf{h}} = \sum_{i=1}^{n} \mathbf{t}_i \frac{\exp\left(\langle \mathbf{t}_i, \mathbf{z} \rangle + b_{j-i} + \mathbf{1}_{\text{masked}}(i)\right)}{\sum_{l=1}^{n} \exp\left(\langle \mathbf{t}_l, \mathbf{z} \rangle + b_{j-l} + \mathbf{1}_{\text{masked}}(l)\right)}, \tag{18}$$

where $\mathbf{1}_{\text{masked}}(i)$ is an indicator function that equals $-\infty$ if word $i$ is masked and zero otherwise. The expression in (18) has the same structure as that adopted in T5 transformer, where the indicator function serves as the mask function to prevent the model from assigning weights to the masked words. In this way, the concepts of word masks and the positional encoding functions are unified by $u$ in (17). Conversely, T5 transformer is a realization of the generalized attention with the preference weights $u$ specified in (17).

**Generalized attention structures suggested by the optimal solution.** While T5 transformer has implicitly adopted the generalized attention, (16) suggests potential for further generalizations. For instance, in T5 transformer, the function that outputs the template's preference weights only considers the word masks and the word's relative positions. This function could also be generalized to factor in the input sentence contexts, and the output weights encode the importance of each word before giving the local information stored in $\mathbf{z}$. The same idea could be applied to the image captioning example to replace the uniform preference weights. By adding a neural network taking the input image to generate non-uniform preference weights, we devise a mechanism to estimate the importance of each part of the image before the caption generation. In this way, the newly added network collects global information from the image to propose a preference distribution, which could be updated locally based on the current generation stage encoded in $\mathbf{z}$.

Besides, although we mainly focus on the case when $u$ is discrete, we want to emphasize that the analysis performed in Section 6 also covers continuous $u$. This hints that a continuous attention mechanism could also be implemented, which might prove to be useful in some applications.

Moreover, our theoretical work enables the design of more general attention structures; for instance, KL-divergence in the optimization problem (3) requires estimated distribution to share support with preference

---

[7]They also simplified the layer normalization (Lei Ba et al., 2016) for faster training and inference speed.

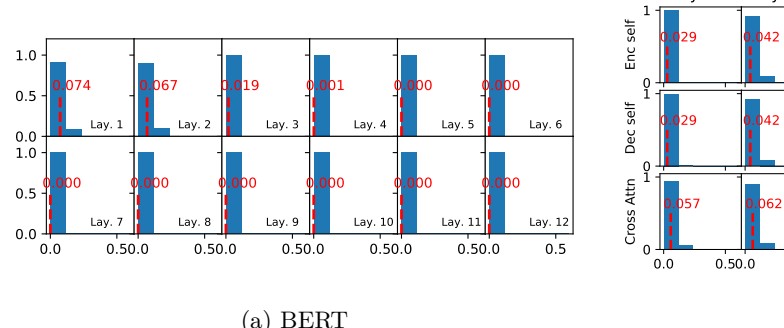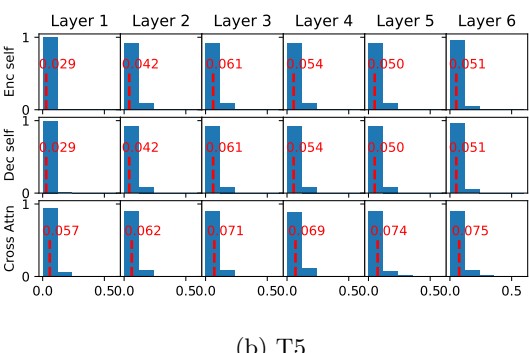

(a) BERT  (b) T5

Figure 4: The distribution of relative deviations $\frac{\|\boldsymbol{\lambda}^* - \alpha\mathbf{z}\|}{\|\boldsymbol{\lambda}^*\|}$ for the attention in BERT and T5. The red vertical lines mark the average of the errors.

distribution, which may not be desired in many tasks. (e.g. translation, where the target should be unaffected if we replace some words in the source sentence with synonyms.) Using our theory, in Sec 9, we show that this can be achieved by replacing KL divergence with an optimal transport (OT)-based measure that handles word similarities in their embedding space.

## 8  Empirical evidence

To show the proposed optimization problem (3) indeed provides a principle justifying the design of attention modules, we show that the maximizer $\boldsymbol{\lambda}^*$ of its dual problem (7) nearly coincides with its approximated counterpart used in the pretrained BERT model (Devlin et al., 2019) and T5-small (Raffel et al., 2020). Verification on other popular attention-based models yielded similar results.

Let $\mathbf{x}_i \in \mathbb{R}^d$ for $i \in 1, 2 \ldots, n$, $\mathbf{y}_j \in \mathbb{R}^d$ for $i \in 1, 2 \ldots, m$ and $K, Q, V \in \mathbb{R}^{d' \times d}$. The $k^{\text{th}}$ outputs of BERT attention and T5 are respectively,

$$\textbf{BERT}: \ \sum_{i=1}^{n} V\mathbf{x}_i \ \frac{\exp\left(\langle K\mathbf{x}_i, Q\mathbf{x}_k\rangle/\sqrt{d'}\right)}{\sum_{j=1}^{n}\exp\left(\langle K\mathbf{x}_j, Q\mathbf{x}_k\rangle/\sqrt{d'}\right)} \qquad \textbf{T5}: \ \sum_{i=1}^{n} V\mathbf{x}_i \ \frac{u_i\exp\left(\langle K\mathbf{x}_i, Q\mathbf{y}_k\rangle/\sqrt{d'}\right)}{\sum_{j=1}^{n}\exp\left(\langle K\mathbf{x}_j, Q\mathbf{y}_k\rangle/\sqrt{d'}\right)}. \quad (19)$$

Here, T5 has three types of attention, self-attentions in the encoder and the decoder and the cross-attention connecting them. For the two self-attentions, $\mathbf{x}_i = \mathbf{y}_i$ and $m = n$.

Following the reparameterization method used by Ramsauer et al. (2021), for BERT, setting $\alpha = 1$, $\mathbf{t}_i = \frac{\mathbf{x}_i}{\sqrt{d'}}$, $\mathbf{z} = K^\top Q\mathbf{x}_k$, $V' = V\sqrt{d'}$, and $u_i \propto 1$ yields $V'\sum_{i=1}^{n} \mathbf{t}_i \ \frac{u_i\exp\langle\mathbf{t}_i,\mathbf{z}\rangle}{\sum_{j=1}^{n}u_j\exp\langle\mathbf{t}_j,\mathbf{z}\rangle}$, where the summation part is the one derived in (16).[8] Likewise, for T5, we use the same setting as BERT except that $u_i$ is computed based on its positional encoding and $\mathbf{z} = K^\top Q\mathbf{y}_k$ for the cross-attention.

We find $\boldsymbol{\lambda}^*$ by plugging $\alpha$, $u_i$'s, $\mathbf{t}_i$'s and $\mathbf{z}$ into (7) followed by performing gradient ascent. We then calculate the relative deviation $\frac{\|\boldsymbol{\lambda}^* - \alpha\mathbf{z}\|}{\|\boldsymbol{\lambda}^*\|}$ of its approximated counterpart $\alpha\mathbf{z}$ and report its distribution in Fig 4 for each attention layer by taking the average over the attention heads. We report the distributions for each head in Appx C. As Fig 4 indicates, $\boldsymbol{\lambda}^*$ almost coincides with its approximated counterpart $\alpha\mathbf{z}$ inferred by BERT and T5. As a result, the T5 and BERT's attention inference can be seen as solving the proposed convex problem, which corroborates that problem (3) gives a principle justifying the design of attention.

---

[8]Templates $\mathbf{t}_i$ absorb the scaling factor $d'^{-\frac{1}{2}}$ so that their norms remain largely unchanged as $d'$ increases. Thus, $u$ has bounded moments, and Theorem 2 applies. Note that it is a common practice to scale outputs before performing theoretical analysis (e.g., see the work of Arora et al. (2019)).

## 9 An optimal transport-based attention

In Sec 7, we mentioned that our theoretical work enables the design of more general attention structures. Let $\mathbb{R}_+$ denote the set of non-negative real numbers. In this section, we provide an example by replacing the KL-divergence in (3) with an entropy-regularized OT-based measure (Cuturi, 2013):

$$\mathcal{W}_\gamma(p, u; M) = \min_{X \in U(p,u)} \langle M, X \rangle - \gamma H(X), \tag{20}$$

where $\gamma > 0$, $H(X) = \sum_{i,j=1}^N -X_{ij} \log X_{ij}$ is the entropy of $X$, $U(p, u) = \{X \in \mathbb{R}_+^{|\mathcal{A}| \times |\mathcal{A}|}; X\mathbf{1} = p, X^T \mathbf{1} = u\}$ and $M \in \mathbb{R}_+^{|\mathcal{A}| \times |\mathcal{A}|}$ is a cost matrix that measures the similarity between each pair of the templates in $\mathcal{A}$.[9] The entropy regularization makes the minimizer $X^*$ in (20) change smoothly in terms of $p, u$ and $M$, which stabilizes and speeds up evaluation of $\mathcal{W}$ (Cuturi, 2013). When $\gamma \to 0$, $M(\mathbf{t}, \mathbf{t}') = d_\mathcal{A}(\mathbf{t}, \mathbf{t}')^\rho$, $\mathcal{W}_\gamma^{1/\rho}$ is reduced to the Wasserstein $\rho$-distance. We note that, due to the entropy term, for fixed $u$ and $M$, the true preference distribution $\tilde{u}$ that minimizes $\mathcal{W}_\gamma(\tilde{u}, u; M)$ is slightly deviated from $u$ and will approach to $u$ if $\gamma \to 0$. (see Appx D for details.) Let $\tilde{\boldsymbol{\mu}}$ denote the expectation of $\tilde{u}$. Then we can rewrite (3) as

$$\min_p \frac{\alpha}{2} \left\| (\tilde{\boldsymbol{\mu}} + \mathbf{z}) - \int_{\mathbb{R}^d} \mathbf{a} p(\mathbf{a}) \, d\mathbf{a} \right\|^2 + \mathcal{W}_\gamma(p, u; M). \tag{21}$$

Following a similar procedure presented in Sec 6 (the derivation is given in Appx D), we can derive and solve its Fenchel dual problem and show that when both $\alpha$ and $\frac{\alpha}{\gamma}$ are small, the minimizer $p^*$ takes the form

$$p^*(\mathbf{t}) = \sum_{i=1}^n u_i \frac{\exp\left( \left( \alpha \mathbf{t}^T \mathbf{z} - M(\mathbf{t}, \mathbf{t}_i) \right) / \gamma \right)}{Z_i} \tag{22}$$

with $Z_i = \sum_{\mathbf{t}' \in \mathcal{A}} \exp\left( \left( \alpha (\mathbf{t}')^T z - M(\mathbf{t}', \mathbf{t}_i) \right) / \gamma \right)$. Substituting (22) into (4), we get the OT-based attention.

**The OT-based attention considers all templates in $\mathcal{A}$.** In comparison to the generalized attention derived in Sec 6, the OT-based one assigns non-zero weights to all templates in $\mathcal{A}$. To see how it works, consider an extreme case in which the templates are partitioned into several groups. If two templates $\mathbf{t}, \mathbf{t}'$ belong to the same group, $M(\mathbf{t}, \mathbf{t}') = 0$; otherwise, $M(\mathbf{t}, \mathbf{t}) = \infty$. Moreover, templates within the same groups are very similar in the sense that their inner products with $\mathbf{z}$ are approximately equal. Suppose $\mathbf{t}_i$ belongs to a group $\mathcal{G}$ and other templates $\mathbf{t}_{j \neq i}$ do not, then for all $\mathbf{t} \in \mathcal{G}$, we have $p^*(\mathbf{t}) = u_i / |\mathcal{G}|$. That is, all templates of $\mathcal{G}$ share the weight of $\mathbf{t}_i$ and thus be potentially trained even if most of them do not appear in the input. In general, if a template $\mathbf{t}$ is similar to some $\mathbf{t}_i \in \mathbf{T}$ (i.e., $M(\mathbf{t}, \mathbf{t}_i)$ is small), it will share $\mathbf{t}_i$'s weight although it does not appear in $\mathbf{T}$. In contrast, for regular attention, only templates in $\mathbf{T}$ can be assigned non-zero weights. The peculiar property of OT-based attention is desired in some practical tasks. For example, in an NLP problem, synonyms intuitively have similar templates. Then, if a word appears in the input sentence and is trained, its synonyms should be trained in a similar way and thus be assigned a similar weight (because replacing a word with its synonym does not alter the input in a semantic sense).

Likewise, in the Vision Transformer (ViT) (Dosovitskiy et al., 2021), images are divided into small patches, each of which is conceptually treated as a word. Consequently, an image composed of these patches is analogous to a sentence. A multilayer transformer, similar to BERT, is then used to extract features from these patches. Finally, a special learnable token is incorporated to aggregate these features (templates) using an attention mechanism, and the aggregated result is fed into a classifier for image classification. Intuitively, images of the same class consist of visually similar patches and replacing patches in an image with visually similar patches should not alter its class. Thus, it is reasonable for the last attention layer to share the template sets for images of the same classes and adopt the OT-based attention to train the templates associated with visually similar patches.

---

[9]A smaller $M_{ij}$ implies a larger similarity between $\mathbf{t}_i$ and $\mathbf{t}_j$. While many OT-related problems define $M$ by embedding templates into a metric space $(\mathcal{A}, d_\mathcal{A})$ with $M(\mathbf{t}, \mathbf{t}') = d_\mathcal{A}(\mathbf{t}, \mathbf{t}')^\rho$, $\rho \geq 1$, our discussion makes no assumption on $M$ other than it is non-negative and symmetric, and $M(\mathbf{t}, \mathbf{t}) < M(\mathbf{t}', \mathbf{t})$ for all $\mathbf{t}' \neq \mathbf{t}$.

To corroborate our claims, we test the ViT and its OT-based variant on Fashion-MNIST (Xiao et al., 2017), CIFAR10 and CIFAR100 (Krizhevsky, 2009). The OT-ViT model is identical to the ViT, except that the final transformer layer is substituted with OT-based attention where $M(\mathbf{a}, \mathbf{b}) = -\mathbf{a}^\top \mathbf{b} + C$, $\alpha = 1$ and $\gamma = \sqrt{\text{hidden dim}}$. ($C$ is an upper bound of all possible template pairs' inner products, which ensures $M$ is nonnegative.) To improve the training efficiency, when training OT-ViT, there is a 50% chance that the set $\mathbf{T}$ consists solely of templates extracted from the input image and a 50% chance that $\mathbf{T}$ also includes templates from another randomly selected image of the same class. During testing, $\mathbf{T}$ consists only of templates from the input image.

| LR | $3 \times 10^{-3}$ | $3 \times 10^{-4}$ | $3 \times 10^{-5}$ |
|---|---|---|---|
| ViT | $0.228 \pm 0.01$ | $\mathbf{0.463} \pm 0.01$ | $0.452 \pm 0.01$ |
| OT-ViT | $0.175 \pm 0.01$ | $\mathbf{0.491} \pm 0.01$ | $0.412 \pm 0.01$ |

Table 1: Test accuracies of ViT and OT-ViT on CIFAR100 with various learning rates (LR).

| | Fashion-MNIST | CIFAR10 | CIFAR100 |
|---|---|---|---|
| ViT | $0.928 \pm 0.01$ | $0.751 \pm 0.01$ | $0.463 \pm 0.01$ |
| OT-ViT | $\mathbf{0.937} \pm 0.01$ | $\mathbf{0.772} \pm 0.02$ | $\mathbf{0.491} \pm 0.01$ |

Table 2: Test accuracies of ViT and OT-ViT on Fashion-MNIST, CIFAR10 and CIFAR100.

Throughout our experiments, we fixed the patch size to be $4 \times 4$ and the dropout rate to be 0.2. To ensure a fair and tractable comparison, we constrained both models to have 3.2M parameters. Under this constraint, we traded off the number of layers and hidden dimensions of the Vision Transformer (ViT) model to achieve the best performance on CIFAR100 (Krizhevsky, 2009). The study showed that a six-layer ViT model with a hidden dimension of 512 had the optimal performance. We then used this setting for both the ViT and OT-ViT models throughout the remaining experiments. (Note that for a fixed hidden dimension, the OT-based attention has a nearly identical number of parameters to the regular transformer implementation.) Similarly, we searched for the optimal learning rate (LR) of both models on CIFAR100 and reported the test accuracy with the 95% confidence intervals in Table 1. The results indicate that both models achieved the best performance when the learning rate was set to $3 \times 10^{-4}$. We, therefore, used this learning rate selection when training the models on the other datasets.

In Table 2, we compare the performances of ViT and OT-ViT on Fashion-MNIST (Xiao et al., 2017), CIFAR10 and CIFAR100 (Krizhevsky, 2009) by reporting their test accuracies with the 95% confidence interval. As demonstrated, OT-ViT consistently outperforms ViT, highlighting the effectiveness of OT-based attention.

## 10 Conclusion

This paper presented a new perspective on understanding the attention mechanism by showing that it can be viewed as a solver of a family of inference tasks. These tasks involve improving the noisy estimate of a distribution $p$'s mean by a preference distribution that encodes some beliefs of $p$'s value. We have used three running examples with the typical model architectures to show that such tasks naturally exist in neural network design. We then abstracted a convex optimization problem from these tasks and derived a closed-form approximation of the optimal solution by solving the problem's Fenchel dual. We find that the closed-form approximation can be seen as a generalized attention layer, and one of its special cases is equivalent to the dot-product attention adopted in transformers. We further analyzed the general form and showed that T5 transformer implicitly adopts the generalized attention structure with attention weights unifying the concepts of the word masks and the positional encoding functions. We empirically show that our framework can well-explain the attention inference in the pretrained BERT and T5 models. To demonstrate the potential for designing more general attention structures, we replaced the KL divergence with an OT-based measure, deriving an OT-based attention structure that removes the support constraints on $p^{(k)}$ mentioned in the examples.

This paper also presents a principled justification for the design of attention modules in neural networks. Specifically, there is a general assumption that because attention in humans narrows the search space, a similar phenomenon is at play in transformers. In this paper, we have shown that the mechanism corresponds to proposing a preference distribution over the templates, followed by adjusting it using a noisy mean shift estimation. The generalized attention structure presented potentially opens the door to a wide design

space. For example, the preference weights need not be derived from the positional encoding functions; they could integrate a variety of information provided by other network components. Additionally, this research successfully demonstrates a novel approach to analyze the functioning of a neural network component, namely, via isolating the component from the complex network structure and asking: is there a "local problem" that is solved by the design of this component?

**Broader impact.** This paper presents a new perspective on understanding attention and derives a generalized attention structure. Our work is foundational, which we believe does not have direct negative societal impacts. Due to the very wide range of applications of attention, such as self-driving (Kim & Canny, 2017), healthcare (Ma et al., 2017) and protein interaction prediction (Tsubaki et al., 2018), we expect our works can facilitate the algorithm developments in these areas, which may have unexpected impacts.

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

# A   Derivation of (11) for preference distributions of bounded moments

Assume a preference distribution $u$ has bounded moments. Then its moment generating function

$$M(\boldsymbol{\lambda}) = \int_{\mathbb{R}^d} \langle \mathbf{a}, \boldsymbol{\lambda} \rangle u(\mathbf{a}) \mathrm{d}\mathbf{a} = 1 + \langle M'(0), \boldsymbol{\lambda} \rangle + \frac{1}{2} \langle \boldsymbol{\lambda}, M''(0) \boldsymbol{\lambda} \rangle + o(\|\boldsymbol{\lambda}\|^2), \tag{23}$$

where

$$M'(0) = \int \mathbf{a} u(\mathbf{a}) \mathrm{d}\mathbf{a} = \boldsymbol{\mu}, \tag{24}$$

$$M''(0) = \int \mathbf{a}\mathbf{a}^\top u(\mathbf{a}) \mathrm{d}\mathbf{a}. \tag{25}$$

Notice that

$$\log(1 + x) = t - \frac{t^2}{2} + \frac{t^3}{3} - \frac{t^4}{4} + \cdots = t - \frac{t^2}{2} + o(t^2). \tag{26}$$

Thus,

$$\begin{aligned}
\log(M(\boldsymbol{\lambda})) &= \left( \langle M'(0), \boldsymbol{\lambda} \rangle + \frac{1}{2} \langle \boldsymbol{\lambda}, M''(0) \boldsymbol{\lambda} \rangle + o(\|\boldsymbol{\lambda}\|^2) \right) \\
&\quad - \frac{1}{2} \left( \langle M'(0), \boldsymbol{\lambda} \rangle + \frac{1}{2} \langle \boldsymbol{\lambda}, M''(0) \boldsymbol{\lambda} \rangle + o(\|\boldsymbol{\lambda}\|^2) \right)^2 \\
&\quad + o\left( \left( \langle M'(0), \boldsymbol{\lambda} \rangle + \frac{1}{2} \langle \boldsymbol{\lambda}, M''(0) \boldsymbol{\lambda} \rangle + o(\|\boldsymbol{\lambda}\|^2) \right)^2 \right) \\
&= \langle M'(0), \boldsymbol{\lambda} \rangle + \frac{1}{2} \left( \langle \boldsymbol{\lambda}, M''(0) \boldsymbol{\lambda} \rangle - \langle M'(0), \boldsymbol{\lambda} \rangle^2 \right) + o\left( \|\boldsymbol{\lambda}\|^2 \right) \\
&= \langle \boldsymbol{\mu}, \boldsymbol{\lambda} \rangle + \frac{1}{2} \boldsymbol{\lambda}^\top \Sigma \boldsymbol{\lambda} + o\left( \|\boldsymbol{\lambda}\|^2 \right),
\end{aligned}$$

where

$$\Sigma = M''(0) - M'(0)M'(0)^\top = \int (\mathbf{a} - \boldsymbol{\mu})(\mathbf{a} - \boldsymbol{\mu})^T u(\mathbf{a}) \mathrm{d}\mathbf{a}.$$

## B    Other perspectives to derive (3)

**A Maximum Likelihood Perspective.** The optimization problem in (3) can be derived using the maximum log likelihood method by treating the KL-divergence term as a regularizer. According to (2), the difference $(\boldsymbol{\mu} + \mathbf{z}) - \mathbf{h}$ follows a Gaussian distribution $\mathcal{N}(\mathbf{0}, \sigma^2 \mathbf{I})$. This implies the log likelihood function $\ell(\mathbf{z}) \propto -\frac{1}{2\sigma^2} \|(\boldsymbol{\mu} + \mathbf{z}) - \mathbf{h}\|^2$. Maximizing it with the KL-divergence term as a regularizer is the same as minimizing

$$\frac{1}{2\sigma^2} \|(\boldsymbol{\mu} + \mathbf{z}) - \mathbf{h}\|^2 + \eta \mathcal{K}(p, u), \tag{27}$$

where $\eta > 0$ controls the strength of the regularization. Substituting (1) into (27) followed by rearrangement yields

$$\min_p \frac{1}{2\eta\sigma^2} \left\| (\boldsymbol{\mu} + \mathbf{z}) - \int_{\mathbb{R}^d} \mathbf{a} p(\mathbf{a}) \, \mathrm{d}\mathbf{a} \right\|^2 + \mathcal{K}(p, u), \tag{28}$$

which is equivalent to (3) by setting $\alpha^{-1} = \eta\sigma^2$.

**A Bayesian perspective.** Given observed data and prior belief about the distribution of parameters, Bayesian inference allows us to update this distribution to reflect the new knowledge. Assume that the distribution $p$ is specified by parameters $\theta$. By considering $\boldsymbol{\mu} + \mathbf{z}$ as the observed data, we will show that picking the $p_\theta$ that minimizes (3) is the same as choosing the $\theta^*$ that maximizes the posterior density of $\theta$ given the observed data.

Let $\vartheta$ be the parameters of the preference distribution $u_\vartheta$ and suppose the prior distribution $f(\theta|\vartheta)$ of $\theta$ satisfies

$$f(\theta|\vartheta) \propto \exp\left( -\eta \mathcal{K}(p_\theta, u_\vartheta) \right), \tag{29}$$

where $\eta > 0$ is a hyper-parameter that controls the decaying speed of the probability density as $p_\theta$ deviates from $u_\vartheta$.

In (2), we have assumed that given $\theta$, $(\boldsymbol{\mu} + \mathbf{z}) - \mathbf{h}_\theta$ follows a spherical Gaussian distribution $\mathcal{N}(\mathbf{0}, \sigma^2 \mathbf{I})$, where $\mathbf{h}_\theta$ is the mean of $p_\theta$. Therefore, given its parameter $\theta$, the probability density function of $\boldsymbol{\mu} + \mathbf{z}$ is

$$f(\boldsymbol{\mu} + \mathbf{z}|\theta) = f(\boldsymbol{\mu} + \mathbf{z}|\mathbf{h}_\theta) \propto \exp\left( -\frac{1}{2\sigma^2} \|(\boldsymbol{\mu} + \mathbf{z}) - \mathbf{h}_\theta\|^2 \right). \tag{30}$$

Then the posterior distribution of $\theta$ satisfies

$$f(\theta|\boldsymbol{\mu} + \mathbf{z}, \vartheta) \propto f(\boldsymbol{\mu} + \mathbf{z}|\theta) \; f(\theta|\vartheta)$$
$$\propto \exp\left( -\frac{1}{2\sigma^2} \|(\boldsymbol{\mu} + \mathbf{z}) - \mathbf{h}_\theta\|^2 - \eta \mathcal{K}(p_\theta, u_\vartheta) \right).$$

Finding $\theta^*$ that maximizes the posterior $f(\theta|\boldsymbol{\mu} + \mathbf{z}, \vartheta)$ is the same as finding

$$p_\theta^* = \underset{p_\theta}{\operatorname{argmin}} \left\{ \frac{1}{2\sigma^2} \|(\boldsymbol{\mu} + \mathbf{z}) - \mathbf{h}_\theta\|^2 + \eta \mathcal{K}(p_\theta, u_\vartheta) \right\}$$
$$= \underset{p_\theta}{\operatorname{argmin}} \left\{ \frac{1}{2\eta\sigma^2} \|(\boldsymbol{\mu} + \mathbf{z}) - \mathbf{h}_\theta\| + \mathcal{K}(p_\theta, u_\vartheta) \right\},$$

which is equivalent to (3) by setting $\alpha^{-1} = \eta\sigma^2$.

## C Extra experimental results

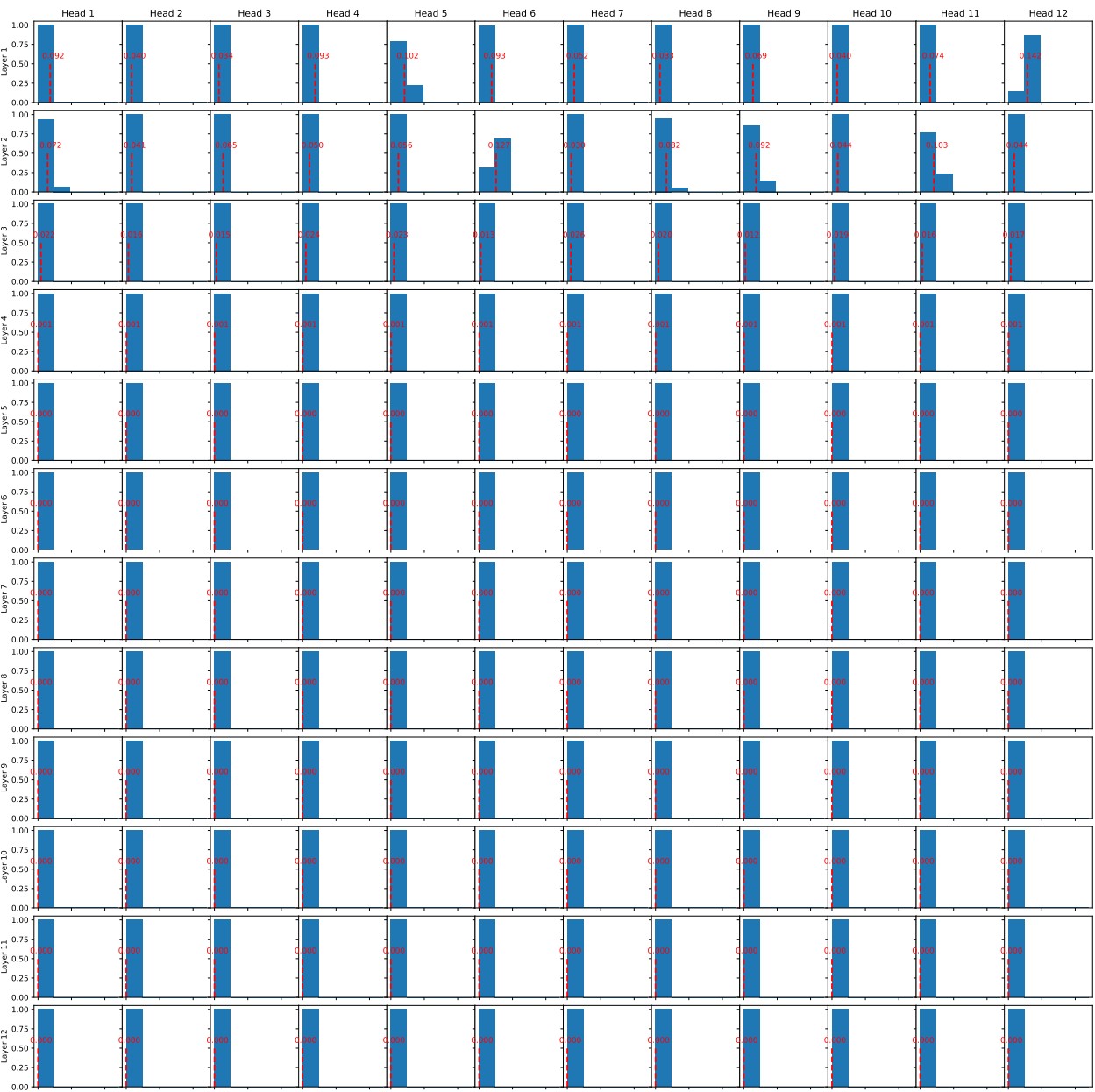

Figure 5: The distribution of relative errors $\frac{\|\boldsymbol{\lambda}^* - \alpha \mathbf{z}\|}{\|\boldsymbol{\lambda}^*\|}$ for the attention in BERT. The red vertical lines mark the average of the errors.

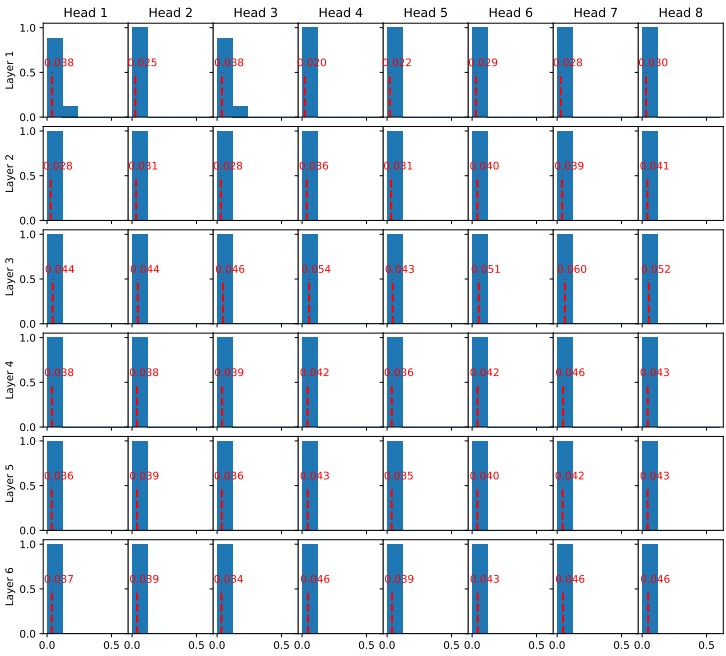

Figure 6: The distribution of relative errors $\frac{\|\boldsymbol{\lambda}^* - \alpha \mathbf{z}\|}{\|\boldsymbol{\lambda}^*\|}$ for the self-attention of the encoder in T5. The red vertical lines mark the average of the errors.

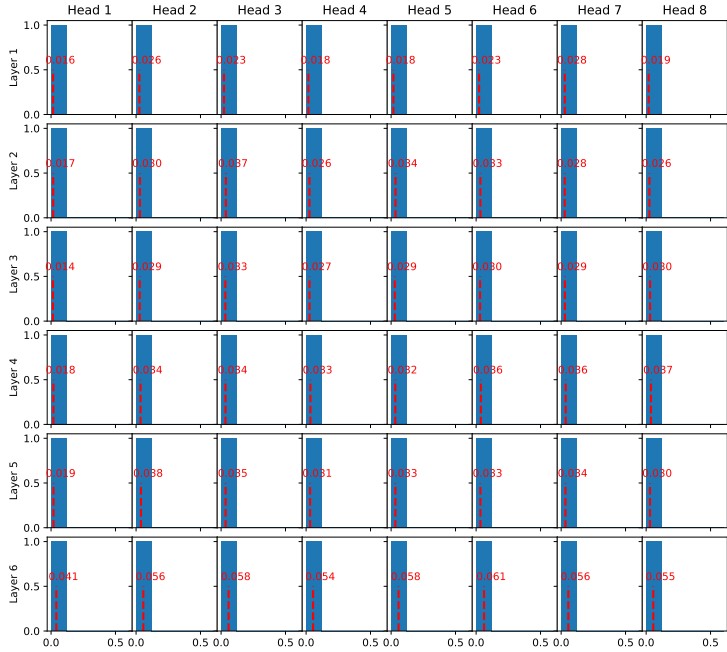

Figure 7: The distribution of relative errors $\frac{\|\boldsymbol{\lambda}^* - \alpha \mathbf{z}\|}{\|\boldsymbol{\lambda}^*\|}$ for the self-attention of the decoder in T5. The red vertical lines mark the average of the errors.

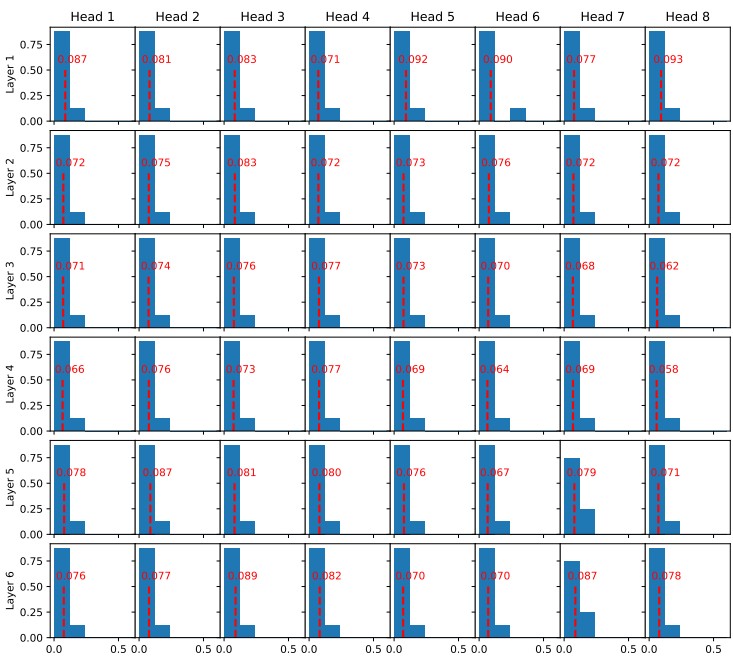

Figure 8: The distribution of relative errors $\frac{\|\boldsymbol{\lambda}^* - \alpha \mathbf{z}\|}{\|\boldsymbol{\lambda}^*\|}$ for the cross-attention in T5. The red vertical lines mark the average of the errors.

## D  Details on the derivation of OT-based attention

According to the discussion in Sec 9, we consider the optimization problem

$$p^* = \underset{p}{\operatorname{argmin}} \frac{\alpha}{2} \left\| (\tilde{\boldsymbol{\mu}} + \mathbf{z}) - \int_{\mathbb{R}^d} \mathbf{a} p(\mathbf{a}) \, d\mathbf{a} \right\|^2 + \mathcal{W}_\gamma(p, u; M) \tag{31}$$

where $\tilde{\boldsymbol{\mu}}$ denotes the mean of the true preference distribution $\tilde{u}$ that minimizes $f(p) = \mathcal{W}_\gamma(p, u; M)$. We will show in Prop 1 that

$$\tilde{\boldsymbol{\mu}} = \sum_{\mathbf{t}, \mathbf{t}' \in \mathcal{A} \times \mathcal{A}} u(\mathbf{t}') \frac{\exp\left(-M(\mathbf{t}, \mathbf{t}')/\gamma\right)}{\sum_{\mathbf{t}'' \in \mathcal{A}} \exp\left(-M(\mathbf{t}'', \mathbf{t}')/\gamma\right)} \mathbf{t}. \tag{32}$$

Cuturi and Peyre Cuturi & Peyre (2016) proved that the Fenchel dual of $\mathcal{W}_\gamma(\mathbf{d}; r, M)$ is

$$\mathcal{W}_\gamma^*(p; u, M) = \gamma \left( H(u) + \sum_{\mathbf{t} \in \mathcal{A}} u(\mathbf{t}) \, \log \left[ \sum_{\mathbf{t}' \in \mathcal{A}} \exp \left( \gamma^{-1} \big( p(\mathbf{t}) - M(\mathbf{t}, \mathbf{t}') \big) \right) \right] \right) \tag{33}$$

for $p \in \mathbb{R}^N$; and, for $\mathbf{t} \in \mathcal{A}$

$$\left[ \nabla_p \mathcal{W}_\gamma^*(p; u, M) \right]_{\mathbf{t}} = \sum_{\mathbf{t}' \in \mathcal{A}} \frac{u(\mathbf{t}') \exp \left( \gamma^{-1} \big( p(\mathbf{t}) - M(\mathbf{t}, \mathbf{t}') \big) \right)}{\sum_{\mathbf{t}'' \in \mathcal{A}} \exp \left( \gamma^{-1} \big( p(\mathbf{t}'') - M(\mathbf{t}', \mathbf{t}'') \big) \right)}, \tag{34}$$

where $\left[ \nabla_p \mathcal{W}_\gamma^*(p; u, M) \right]_{\mathbf{t}}$ denote the entry in $\left[ \nabla_p \mathcal{W}_\gamma^*(p; u, M) \right]$ that is associated to template $\mathbf{t}$. By the Fenchel's duality theorem, we know that $p^*$ in (31) takes the form

$$p^*(\mathbf{t}) = \sum_{\mathbf{t}' \in \mathcal{A}} \frac{u(\mathbf{t}') \exp \left( \gamma^{-1} \big( \mathbf{t}^\top \lambda^* - M(\mathbf{t}, \mathbf{t}') \big) \right)}{\sum_{\mathbf{t}'' \in \mathcal{A}} \exp \left( \gamma^{-1} \big( (\mathbf{t}'')^\top \lambda^* - M(\mathbf{t}', \mathbf{t}'') \big) \right)}, \tag{35}$$

where

$$\lambda^* = \arg\max_{\lambda \in \mathbb{R}^d} \langle \tilde{\boldsymbol{\mu}} + \mathbf{z}, \lambda \rangle - \frac{1}{2\alpha} \|\lambda\|^2 - \mathcal{W}_\gamma^* \left( [\mathbf{t}^T \lambda | \mathbf{t} \in \mathcal{A}]; u, M \right)$$

$$= \arg\max_{\lambda \in \mathbb{R}^d} \langle \tilde{\boldsymbol{\mu}} + \mathbf{z}, \lambda \rangle - \frac{1}{2\alpha} \|\lambda\|^2 - \gamma \sum_{\mathbf{t} \in \mathcal{A}} u(\mathbf{t}) \ \log \left[ \sum_{\mathbf{t}' \in \mathcal{A}} \exp \left( \frac{(\mathbf{t}')^\top \lambda - M(\mathbf{t}, \mathbf{t}')}{\gamma} \right) \right]. \tag{36}$$

**The true preference distribution.** The Fenchel dual perspective allows us to derive a closed-form expression for the minimizer of $f(p) = \mathcal{W}_\gamma(p, u; M)$, which we refer as the true preference distribution $\tilde{u}$ in the main text. We will also show that $\tilde{u}$ approaches to the preference $u$ as $\gamma \to 0$.

Notice that, by definition, $\tilde{u} \to p^*$ when $\alpha \to 0$ in (31). In this case, the optimization of $\lambda$ in (36) gets an infinite penalty on its L2 norm and thus $\|\lambda^*\|^2 = 0$. Therefore, we have

**Proposition 1.** $\mathcal{W}_\gamma(p; u, M)$ *has the minimizer* $\tilde{u}(\mathbf{t})$ *taking the form*

$$\tilde{u}(\mathbf{t}) = \sum_{\mathbf{t}' \in \mathcal{A}} u(\mathbf{t}') \frac{\exp\left(-M(\mathbf{t}, \mathbf{t}')/\gamma\right)}{\sum_{\mathbf{t}'' \in \mathcal{A}} \exp\left(-M(\mathbf{t}', \mathbf{t}'')/\gamma\right)}, \tag{37}$$

*for* $\mathbf{t} \in \mathcal{A}$. *Besides, its mean*

$$\tilde{\boldsymbol{\mu}} = \sum_{\mathbf{t}, \mathbf{t}' \in \mathcal{A} \times \mathcal{A}} u(\mathbf{t}') \ \frac{\exp\left(-M(\mathbf{t}, \mathbf{t}')/\gamma\right)}{\sum_{\mathbf{t}'' \in \mathcal{A}} \exp\left(-M(\mathbf{t}'', \mathbf{t}')/\gamma\right)} \mathbf{t}. \tag{38}$$

When $\gamma \to 0$, $\frac{\exp\left(-M(\mathbf{t}, \mathbf{t}')/\gamma\right)}{\sum_{\mathbf{t}'' \in \mathcal{A}} \exp(-M(\mathbf{t}', \mathbf{t}'')/\gamma)}$ approaches to 1 if $\mathbf{t} = \mathbf{t}'$ and 0 otherwise. Therefore, $\tilde{u}(\mathbf{t}) \to u(\mathbf{t})$ for all $\mathbf{t} \in \mathcal{A}$.

**The derivation of** (22). Then we show how to derive (22) when $\alpha$ and $\frac{\alpha}{\gamma}$ are assumed small.

Within the summation term of (36), for a fixed $\mathbf{t}$

$$\log \left[ \sum_{\mathbf{t}' \in \mathcal{A}} \exp \left( \frac{(\mathbf{t}')^\top \lambda - M(\mathbf{t}, \mathbf{t}')}{\gamma} \right) \right] = \log \left[ \sum_{\mathbf{t}' \in \mathcal{A}} \exp \left( \frac{-M(\mathbf{t}, \mathbf{t}')}{\gamma} \right) \exp \left( \frac{(\mathbf{t}')^\top \lambda}{\gamma} \right) \right]$$

$$= \log \left[ \sum_{\mathbf{t}' \in \mathcal{A}} q_{\mathbf{t}}(\mathbf{t}') Z(\mathbf{t}) \ \exp \left( \frac{(\mathbf{t}')^\top \lambda}{\gamma} \right) \right] = \log \left[ \sum_{\mathbf{t}' \in \mathcal{A}} q_{\mathbf{t}}(\mathbf{t}') \ \exp \left( \frac{(\mathbf{t}')^\top \lambda}{\gamma} \right) \right] + \log Z(\mathbf{t})$$

$$= \log \mathbf{M}_{\mathbf{t}}(\lambda/\gamma) + \log Z(\mathbf{t}), \tag{39}$$

where

$$q_{\mathbf{t}}(\mathbf{t}') = \exp \left( \frac{-M(\mathbf{t}, \mathbf{t}')}{\gamma} \right) \Big/ Z(\mathbf{t}),$$

$$Z(\mathbf{t}) = \sum_{\mathbf{t}' \in \mathcal{A}} \exp \left( \frac{-M(\mathbf{t}, \mathbf{t}')}{\gamma} \right),$$

and $\mathcal{M}_{\mathbf{t}}$ denotes the MGF of $q_{\mathbf{t}}$.

Note that $\log \mathcal{M}_{\mathbf{t}}(\lambda/\gamma)$ is called the cumulant of $q_{\mathbf{t}}$ and has the expansion

$$\log \mathcal{M}_{\mathbf{t}}(\lambda/\gamma) = \boldsymbol{\mu}_{\mathbf{t}}^\top (\lambda/\gamma) + \frac{1}{2} \ (\lambda/\gamma)^\top \ \Sigma_{\mathbf{t}} \ (\lambda/\gamma) + \mathcal{O}(\|\lambda/\gamma\|^3), \tag{40}$$

where

$$\boldsymbol{\mu}_{\mathbf{t}} = \sum_{\mathbf{t}' \in \mathcal{A}} q_{\mathbf{t}}(\mathbf{t}') \ \mathbf{t}' \tag{41}$$

and

$$\Sigma_{\mathbf{t}} = \sum_{\mathbf{t} \in \mathcal{A}} q_{\mathbf{t}}(\mathbf{t}') \ (\mathbf{t}' - \mu_{\mathbf{t}})(\mathbf{t}' - \mu_{\mathbf{t}})^\top \tag{42}$$

respectively denote the mean and the variance-covariance matrix of $q_{\mathbf{t}}$.

Substituting (39) and (40) into (36) yields

$$\lambda^* = \arg\max_{\lambda \in \mathbb{R}^d} \langle \tilde{\boldsymbol{\mu}} + \mathbf{z}, \lambda \rangle - \frac{1}{2\alpha} \|\lambda\|^2$$
$$- \gamma \left[ \left( \sum_{\mathbf{t} \in \mathcal{A}} u(\mathbf{t})\boldsymbol{\mu}_{\mathbf{t}} \right)^\top (\lambda/\gamma) + \frac{1}{2} \sum_{\mathbf{t} \in \mathcal{A}} u(\mathbf{t}) \left( (\lambda/\gamma)^\top \Sigma_{\mathbf{t}}(\lambda/\gamma) \right) + \mathcal{O}(\|\lambda/\gamma\|^3) + \sum_{\mathbf{t} \in \mathcal{A}} u(\mathbf{t}) \log Z(\mathbf{t}) \right]$$
$$= \arg\max_{\lambda \in \mathbb{R}^d} \langle \tilde{\boldsymbol{\mu}} + \mathbf{z}, \lambda \rangle - \frac{1}{2\alpha} \|\lambda\|^2$$
$$- \gamma \left[ (\sum_{\mathbf{t} \in \mathcal{A}} u(\mathbf{t})\boldsymbol{\mu}_{\mathbf{t}})^\top (\lambda/\gamma) + \frac{1}{2} \sum_{\mathbf{t} \in \mathcal{A}} u(\mathbf{t}) \left( (\lambda/\gamma)^T \Sigma_{\mathbf{t}}(\lambda/\gamma) \right) + \mathcal{O}(\|\lambda/\gamma\|^3) \right]$$
$$= \arg\max_{\lambda \in \mathbb{R}^d} \langle \tilde{\boldsymbol{\mu}} + \mathbf{z}, \lambda \rangle - \frac{1}{2\alpha} \|\lambda\|^2$$
$$- \left[ (\sum_i u(\mathbf{t})\boldsymbol{\mu}_{\mathbf{t}})^\top \lambda + \frac{1}{2\gamma} \sum_{\mathbf{t} \in \mathcal{A}} u(\mathbf{t}) \left( \lambda^\top \Sigma_{\mathbf{t}} \lambda \right) + \gamma \mathcal{O}(\|\lambda/\gamma\|^3) \right].$$

When $\alpha$ is assumed to be small, the optimization of $\lambda$ gets a large penalty on its L2 norm and thus, $\|\lambda^*\|^2$ is close to zero. So we have

$$\lambda^* \approx \arg\max_{\lambda \in \mathbb{R}^d} \langle \tilde{\boldsymbol{\mu}} + \mathbf{z}, \lambda \rangle - \frac{1}{2\alpha} \|\lambda\|^2$$
$$- \left[ \left( \sum_{\mathbf{t} \in \mathcal{A}} u(\mathbf{t})\boldsymbol{\mu}_{\mathbf{t}} \right)^\top \lambda + \frac{1}{2\gamma} \sum_{\mathbf{t} \in \mathcal{A}} u(\mathbf{t}) \left( \lambda^\top \Sigma_{\mathbf{t}} \lambda \right) \right]$$

Taking the derivative in terms of $\lambda$ and setting it to zero yields

$$(\tilde{\boldsymbol{\mu}} + \mathbf{z}) - \frac{1}{\alpha}\lambda^* - \sum_{\mathbf{t} \in \mathcal{A}} u(\mathbf{t})\boldsymbol{\mu}_{\mathbf{t}} - \frac{1}{\gamma} \sum_{\mathbf{t} \in \mathcal{A}} u(\mathbf{t})\Sigma_{\mathbf{t}}\lambda^* = 0.$$

As

$$\sum_{\mathbf{t} \in \mathcal{A}} u(\mathbf{t})\boldsymbol{\mu}_{\mathbf{t}} = \sum_{i=1}^{N} u(\mathbf{t}) \sum_{\mathbf{t}' \in \mathcal{A}} q_{\mathbf{t}}(\mathbf{t}')\mathbf{t}' = \sum_{\mathbf{t},\mathbf{t}' \in \mathcal{A} \times \mathcal{A}} u(\mathbf{t}) \frac{\exp\left(-M(\mathbf{t},\mathbf{t}')/\gamma\right)}{\sum_{\mathbf{t}'' \in \mathcal{A}} \exp\left(-M(\mathbf{t},\mathbf{t}'')/\gamma\right)}\mathbf{t}' = \tilde{\boldsymbol{\mu}}, \tag{43}$$

we also have

$$\mathbf{z} - \left( \frac{1}{\alpha}I_d + \frac{1}{\gamma} \sum_{\mathbf{t} \in \mathcal{A}} u(\mathbf{t})\Sigma_{\mathbf{t}} \right) \lambda^* = 0.$$

That is,

$$\lambda^* = \left( \frac{1}{\alpha}I_d + \frac{1}{\gamma} \sum_{\mathbf{t} \in \mathcal{A}} u(\mathbf{t})\Sigma_{\mathbf{t}} \right)^{-1} \mathbf{z}$$
$$= \left( I_d + \frac{\alpha}{\gamma} \sum_{\mathbf{t} \in \mathcal{A}} u(\mathbf{t})\Sigma_{\mathbf{t}} \right)^{-1} (\alpha\mathbf{z}).$$

When $\frac{\alpha}{\gamma}$ is small, the expression becomes simply

$$\lambda^* = \alpha\mathbf{z}.$$

Plugging it into (35), we get

$$p^*(\mathbf{t}) = \sum_{\mathbf{t}' \in \mathcal{A}} \frac{u(\mathbf{t}') \exp\left(\gamma^{-1}\big(\alpha \mathbf{t}^\top \mathbf{z} - M(\mathbf{t}, \mathbf{t}')\big)\right)}{\sum_{\mathbf{t}'' \in \mathcal{A}} \exp\left(\gamma^{-1}\big(\alpha(\mathbf{t}'')^\top \mathbf{z} - M(\mathbf{t}', \mathbf{t}'')\big)\right)}, \tag{44}$$

which is (22).

