# OpenReview forum: "Deciphering Attention Mechanisms: Optimization and Fenchel Dual Solutions"
_TMLR — Rejected by TMLR_

### Review · Reviewer_8TaG · 2024-07-02

**Summary Of Contributions:**

This paper discusses the attention mechanism; namely, given some template (i.e., key-value) vectors $\mathbf{t}_1,\ldots,\mathbf{t}_M$, and an evidence (i.e., query) vector $\mathbf{z}$, the attention $\mathbf{h}$ defined as

$$
\mathbf{h}:=\sum_{i=1}^{M}a_i\mathbf{t}_i
$$

where

$$
a_i=\frac{\exp(\mathbf{t}_i\cdot\mathbf{z})}{\sum_j\exp(\mathbf{t}_j\cdot\mathbf{z})}.
$$

Intuitively, the attention $\mathbf{h}$ can be understood as a convex combination of $\mathbf{t}_1,\ldots,\mathbf{t}_M$ that is "closest to" $\mathbf{z}$; this paper formalizes this intuition. Formally, it shows that $\mathbf{h}$ is approximately the solution of

$$
\min\frac{1}{2}\lVert (\mathbf{\mu}+\mathbf{z})- \mathbf{h}\rVert^2 + KL(a, u),
$$

where $\mathbf{\mu}$ is the mean of $\mathbf{t}_1,\ldots,\mathbf{t}_M$, the $KL(\bullet)$ is the KL-divergence, $a=(a_1,\ldots,a_M)$ is the attention probabilities on $(1,\ldots,M)$ and $u$ is the uniform distribution on $(1,\ldots,M)$.

I said "approximately", because more precisely one should consider the optimization problem

$$
\min\frac{\alpha}{2}\lVert (\mathbf{\mu}+\mathbf{z})- \mathbf{h}\rVert^2 + KL(a, u),
$$

where $\alpha>0$ is a hyper-parameter; the authors show that this problem can be solved by

$$
a_i=\frac{\exp(\mathbf{t}_i\cdot\mathbf{\lambda})}{\sum_j\exp(\mathbf{t}_j\cdot\mathbf{\lambda})}
$$

where $\mathbf{\lambda}=\alpha\mathbf{z}$, when $\alpha$ is sufficiently small.

Then, empirical experiments on T5 and BERT show that the approximation is good enough at $\alpha=1$.

**Audience:**

Yes

**Broader Impact Concerns:**

None.

**Claims And Evidence:**

Yes

**Requested Changes:**

To find at least one application of the formalization, that can naturally generalize beyond the vanilla attention mechanism, and empirically verify its significance.

**Strengths And Weaknesses:**

Strengths:

I like the way that this work formalizes a pertaining intuition of the attention mechanism. The formalization is new to me; I have learned something by reading the paper.

Weaknesses:

I think the authors still need to answer the "so what?" question in order to publish this work. Why does this formalization matter? What can it be useful to?

Currently, the authors claim that the formalization is a justification for the design of attention modules in neural networks. I feel this is a bit overreaching. The attention mechanism is justified by its wide adoption, its simplicity and flexibility in being applicable to various situations, and its good performance. Specific to the Transformer model, attention is critical because it models the correlation among tokens in a sequence. I don't see the formalization of this paper can be connected to such strengths of attention in any significant way.

One potentially fruitful direction, in my opinion, is that the formalization can be naturally generalized beyond the discrete space $(1,\ldots,M)$; i.e., besides attending to a fixed number of vectors $\mathbf{t}_1,\ldots,\mathbf{t}_M$, one may attend to some continuous distribution of vectors with some prior preference, or attend to some structured space. It would be excellent if the authors could find an application where such generalization is natural and can empirically outperform the vanilla attention significantly.

---

> ### Author Response · Authors · 2024-07-27
> **Responses**
>
> Thank you very much for reviewing our submission and the constructive comments.
>
> In the revised version, we have followed your suggestions and implemented OT-based attention to enhance the performance of the vision transformer (ViT). Our empirical results demonstrate that this modification consistently improves ViT, corroborating the effectiveness of the proposed framework.

---

> ### Comment · Reviewer_8TaG · 2024-08-02
> **Thanks for the response**
>
> It is good to see the experiment results; I believe it is toward a promising direction and I really appreciate the work that the authors put into this paper.
>
> That said, I do think that the paper still needs quite a lot of revision. First, it might be helpful to change some terminology: for example, instead of "template" and "evidence", using "key-value vectors" and "query vectors" might just be easier to reach the audience; Second, the lengthy discussion in Section 3 which tries to motivate attention mechanism as a design problem didn't really convince me, and I think this is the case for other reviewers as well; it might be better to shrink this part and devote more space to the experiment settings.
>
> I do find that there are some non-trivial new points in the math of this paper, as I described in my review; but the fact that I get the point does not mean other readers will (as I said, I found the math interesting because it formalizes an intuition that the attention vector is a convex combination of key-value vectors closest to the query in some sense; but the story the authors convey in the paper is rather obscure). Therefore, I believe a major revision of the story is desired.
>
> Also, the new experiment with OT-based attention is a bit different from what I expected. When I say "the formalization can be naturally generalized beyond the discrete space", I expected a "natural generalization", which means **the proposed method should be the same as the canonical attention mechanism when restricted to discrete space**, but one faces difficulty when one tries to naively apply attention to the problem; then, you could say that the math framework in this paper naturally solve the problem, which brings value. (In order to do that I think you need some more math and a good problem to solve.)
>
> However, the proposed OT-based attention replaces the KL-divergence with an entropy-regularized OT-based measure -- while most of the paper tries to justify attention, at the end it proposes to change it to a different thing. Why should I do that? It's not natural at all. I'm not saying you shouldn't propose this new method; but if you want to go that path, the story changes a bit: Now you have to justify, why should one make this change? And the proposed method, as a result, is just to modify the attention score by some similarity matrix, which is not new at all, and definitely not something that can only be achieved with the help of the math framework in this paper. Now you have to compare with other methods that bring information from similar examples during training, such as Mixup (Zhang et al. mixup: Beyond Empirical Risk Minimization); basically there is still a long way to go.

---

> > ### Author Response · Authors · 2024-08-08
> > **Thank you very much for your detailed comments.**
> >
> > Dear Reviewer,
> >
> > Thank you for the replies. To accommodate your suggestions, we have shrunk Sec 3 and used the space to provide additional details on the empirical study in Sec 9. We are leaning toward keeping the terminology "template" and "evidence" as we feel it sounds more intuitive when discussing the motivating examples.
> >
> > Regarding the OT-based attention, we would like to clarify our main message. We argue that there are many problems in machine learning that can be formulated as a mean estimation problem. Specifically, in this problem, we have an unreliable guess of the mean and some ideas of the distribution that can be encoded by a prototype distribution. The challenge is to design an optimization problem that combines these two pieces of information to obtain a better estimation of the mean.
> >
> > In Section 4, we proposed that to solve this problem, we need to find a distribution that is similar to the prototype distribution while having the mean close to the given noisy mean estimation. More importantly, we argue that **the attention mechanism is involved in solving this problem to perform mean prediction.** (Note that the attention mechanism we refer to in this submission is the pipeline to solve this type of problem, rather than the exact transformer structure used in BERT and T5.)
> >
> > Following this philosophy, we can choose different functions to measure the closeness between distributions and the ones between means. When using KL divergence, we recover the T5 and BERT's attention structure. When adopting entropy-regularized OT, we obtain OT-based attention.
> >
> > To support our claims, we empirically show that the approximations we assumed when deriving the regular attention structure are indeed valid for the pretrained T5 and BERT, which in turn corroborates our claim. Additionally, we demonstrate that when we need the attention mechanism to consider the templates in a dictionary but not in the prototype distribution, we can use entropy-regularized OT to measure the distribution's discrepancy. In Sec 9, we show that this approach can improve the performance of ViT when jointly considering the features of the images in the same class during the training (as these features are relevant to the class embedding of that very class but are not in the support of the prototype distribution).
> >
> > (We have rephrased the leading paragraph of Sec 3 to explain the pipeline of our paper better.)
> >
> > It is important to note that our goal here is to provide evidence to support the philosophy that yields attention structure rather than to create new methods for mixing up samples for additional regularization on neural networks. While we are willing to discuss this issue further, we do not think that implementing extra comparisons between the OT-based attention and Mixup-related methods will help support our point.
> >
> > (We apologize for the delayed response as we are currently handling multiple rebuttals simultaneously.)

---

### Review · Reviewer_houp · 2024-07-05

**Summary Of Contributions:**

The authors present an optimization problem central to many estimation tasks and show that a closed-form approximation of its Fenchel dual results in a generalized attention framework. The authors suggest that the success of the T5 architecture is because it has implicitly adopted the general form of the solution. To show the proposed optimization problem provides a principle justifying the design of attention modules, they show the maximizer of its dual problem nearly coincides with the output of the attention block in BERT and T5. The authors use the generalized attention framework to propose an optimal transport-based attention. However, there are no experiments with the proposed method.

**Audience:**

Yes

**Claims And Evidence:**

Yes

**Requested Changes:**

* The presentation in Sec. 4 needs cleaning up. The equations in the *A Maximum Entropy on the Mean Perspective* section are redundant with Eq. (4).
* The lack of experiments is the biggest weakness of this paper and why I think the claims made in the paper are not supported by evidence. The paper should include experiments demonstrating the impact of further generalizations to the T5 architecture or the proposed optimal transport-based attention. As is, I do not consider the proposed optimal transport-based attention a meaningful contribution of the paper.

Requested Clarifications:
* Looking at Eq. (2) from Rioux et al. (2020), shouldn’t $\mathcal{K}(p, u)$ denote the KL divergence from $u$ to $p$? Not from $p$ to $u$?
* Would it be helpful if the number of bins in the histograms in Fig. 4 were increased? As is, these histograms do not show much detail.

**Strengths And Weaknesses:**

Strengths:
* The generalized attention framework is likely of interest to the TMLR audience.
* The authors clearly explain that to solve the optimization problem, the problem must be reformulated as its Fenchel dual, solved, and converted back to one for the original problem.
* Fig. 4 shows the maximizer of its dual problem nearly coincides with the output of the attention block in BERT and T5 (it would be nice to see a naive baseline in these plots).

Weaknesses:
* The authors suggest potential further generalizations of the T5 architecture that arise from Eq. (18), however, there are no experiments showing the impact of these further generalizations.
* A reviewer previously criticized the paper for lacking sufficient insight on how the proposed framework could be used to design improved attention mechanisms. In response to this, the authors proposed an optimal transport-based attention. However, there are no experiments with the proposed method.

---

> ### Author Response · Authors · 2024-07-27
> **Responses**
>
> Thank you very much for your comments and your time to reviewing our submission. Here are our responses:
>
> **Q1**. The authors suggest potential further generalizations of the T5 architecture that arise from Eq. (18), however, there are no experiments showing the impact of these further generalizations.
>
> **A1**. We have added additional empirical results based on OT-based attention in the revised version. The results show that OT-based attention can consistently improve the vision transformer's performance.
>
>
> **Q2**. Looking at Eq. (2) from Rioux et al. (2020), shouldn’t  $\mathcal{K}(p,u)$ denote the KL divergence from $u$ to $p$? Not from   $p$ to  $u$?
>
> **A2**. The relative entropy from $u$ to $p$, denoted by $\mathcal{K}(p, u)$, is also known as the KL divergence of $p$ from $u$. In our revised manuscript, we consistently refer to it as the KL divergence of $p$ from $u$.
>
>
> **Q3**. Would it be helpful if the number of bins in the histograms in Fig. 4 were increased? As is, these histograms do not show much detail.
>
> **A3**. we also tried other options, including doubling the number of bins. We noticed that even in this case, the distribution looks similar as many heads have errors very close to zero, while the bins become too narrow to be rendered properly. Therefore, we chose to report the average error bar as well. We hope the message it conveys is clear: the error is close to zero most of the time.
>
> **Q4**. The presentation in Sec. 4 needs cleaning up. The equations in the A Maximum Entropy on the Mean Perspective section are redundant with Eq. (4).
>
> **A4**. We are unsure if you are referring to the equation below Eq (6). If so, we would like to point out that while the equivalence is somewhat recognized in the convex optimization field, it is not immediately apparent. We believe that the superficial differences between them can offer a distinct perspective for understanding our proposed problem. We are open to further discussion if you could provide more details.

---

> > ### Comment · Reviewer_houp · 2024-08-05
> > **Response to Authors**
> >
> > Thank you for your response and clarifications. Given the added experiments, I think the claims made in the paper are supported by evidence. I will update my review accordingly. However, the paper should include more details describing the training process (e.g., validation set size, preprocessing details, hyperparameter search space).
> >
> > For the equation below Eq (6), I think the relationship between the constrained optimization problem and Eq (4) is well known (similar to the maximum likelihood and maximum a-posterior perspectives in Appx B). If you think the constrained optimization problem offers a distinct perspective for understanding the proposed problem then consider highlighting this more in the text.

---

> ### Author Response · Authors · 2024-08-08
> **Thank you for the reply.**
>
> Dear Reviewer,
>
> Thank you very much for the positive comments! In the revision,
>
> 1) We have added additional details on tuning the parameters and learning rate in Section 9.
> 2) We shrunk the discussion on "A Maximum Entropy on the Mean Perspective." The equation that was originally below Eq (6) has been merged into Eq (6) in the revision.  (It now becomes the one below Eq (4).) We hope these changes will make our discussion better balanced among readers from different backgrounds.

---

### Review · Reviewer_P5hp · 2024-07-16

**Summary Of Contributions:**

This paper tries to deepen the theoretical understanding of the popular attention mechanism, by relating it to a general estimation problem formulation.
The paper derives a (Fenchel) dual formulation of the original estimation problem, and shows that after some approximation steps, the solution can be derived in closed form. In that case, the solution resembles the classical attention module. Numerical experiments show that the approximation steps are reasonable to assume for real-world models.

**Audience:**

Yes

**Broader Impact Concerns:**

None.

**Claims And Evidence:**

No

**Requested Changes:**

* Please address the main weaknesses that are mentioned in the previous part (discuss more realistic attention architecture, explain the relationship between BERT/T5 and the model of section 6, and verify numerically that these models are accurate; discuss how the presented framework provides new understanding).

* I am not sure what the purpose of Section 5 is and how it is important for the rest of the paper.

* Regarding Section 3.1 and 3.2, it does not become really clear what the meaning of z,t,u are n the context of the examples. Very late, the specific formulae are given for BERT and T5, but no interpretation or motivation for this choice is given, other than to match formula (18).

**Strengths And Weaknesses:**

The paper in general tries to keep notation simple in order to be easy to follow. However, sometimes this leads to parts that are hard to understand, for example

"That is, even if a word is absent from the source sentence, it will still be optimized if it has a similar embedding in T(k)."

How can a word be optimized?

Or: what kind of mathematical objects are $\mathcal{A}$ and $a$?

Main Weaknesses:

1) The presented model to connect attention to a certain optimization problem does not encompass multi-head attention, skip connections, or the fact that many attention layers are stacked after each other. Both practical examples (BERT and T5) use at least some of the mentioned techniques (if not all), thus are not matching the presented model closely. This is briefly adressed on page 9, but given that modern transformer architectures are much more complicated than a single attention layer, I would ask the authors to explain in more detail why it would be sufficient to only focus on one attention layer, one head, no skip connection etc.


2) Regarding the experiments, if I understood correctly then they only confirm that the actual solution $\lambda^*$ is close to the approximated solution $\alpha z$. However, this only confirms the arguments of Section 6, which however is simply using prior work and a few approximation steps. In my understanding the main contribution of this paper would be to connect the optimization problem in section 6 to attention/transformer architectures, but empirical validation of this model (e.g. for BERT and T5 on page 11 that is, the choice of z,t,u, etc.) is missing.


3) The connection between (18) and the models for BERT and T5 on page 11 seems to be very loose, which raises the question whether the claimed connection is very specific to attention at all. To be more specific: first, with your choice the value matrix V can not be cast into (18), which weakens the claim of the connection. Second, if we choose $\alpha=1/d$ and $t_i=x_i$ instead, we obtain also BERT. This raises several question that are not addressed in the paper: what does it mean if (18) fits to BERT/T5 with several choices? In particular, the argument of Section 6 relies on $\alpha$ being small, but then what interpretation does $\alpha$ even have for BERT/T5?

**The above two points are the main reason I marked Claims and Evidence as No, but I am open to change my evaluation after discussion/revision.**


Attention is a very well-studied mechanism and multiple interpretations of it have been proposed. It would improve the paper if a short literature overview of the models/interpretations of attention is given, in order to better understand how this paper can offer a new viewpoint. Based on this, it remains unclear to me why the presented viewpoint would be more helpful to understand attention or improve its design compared to previously offered explanations.

---

> ### Author Response · Authors · 2024-07-27
> **Responses (Part 1)**
>
> Thank you for your comments. Our responses aim to clarify the ideas presented in our paper and explain how we have addressed the issues in the revised version.
>
> **Q1**. The presented model to connect attention to a certain optimization problem does not encompass multi-head attention, skip connections, or the fact that many attention layers are stacked after each other. Both practical examples (BERT and T5) use at least some of the mentioned techniques (if not all), thus are not matching the presented model closely.
>
> **A1**: Our submission aims to explore the underlying inductive bias of attention mechanisms. However, we do not claim that our work comprehensively covers all aspects of models involving attention structures. In Section 7, we have included an additional discussion on multihead attention. Regarding skip connections, we treat them similarly to other components of neural networks, as illustrated in Figure 1. Since our primary focus is on justifying the design of attention structures and using these insights to derive general attention frameworks, we believe that actively including residual links would not significantly enhance our work.
>
> **Q2**. With your choice the value matrix V can not be cast into (18), which weakens the claim of the connection.
>
> **A2**. We are uncertain if you are referring to an additional linear transformation, V', in the recovered structure. If so, we would like to clarify that this can be understood as the attention mechanism in practice incorporating an additional linear transformation into the solution of the convex problem. This transformation can be considered as additional components outside our primary focus. We include them in our formula to enhance clarity and coherence in the presentation and connection.
>
> **Q3**. Regarding the choices of $\alpha$ and $t$:  if we choose $\alpha = 1/d$ and $t_i = x_i$ instead, we obtain ...
>
> **A3**. As mentioned in footnote 10, we chose $t_i = x_i / \sqrt {d'}$ because this ensures the norm is bounded as $d'$ increases. To see this, we note that the input $x_i$ is normalized (either by prepossessing or some normalized) with its entries, say in  [-1, 1]. Then it is easy to show its norm is not greater than $\sqrt{d'}$. By choosing $t = x_i / \sqrt {d'}$, the norm of $t$ is bounded by 1. This is important because our analysis assumes finite moments of a distribution. Back to the question, if $t_i = x_i$, then $t_i$ is not bounded; likewise, if $t_i = x_i/d$, then its norm will shrink to $0$ as $d'$ increases. As a result, we chose $t_i = x_i / \sqrt {d'}$.
>
> Regarding the choice of $\alpha$, and $z$,  there could be multiple pairs of $\alpha$ and $z$ that may lead to the same attention architecture. All we want to show here is by choosing an appropriate pair without intentional playing around, our framework can effectively be used to explain how the attention works, which requires $\alpha z$ is close to $\lambda^\star$. In our submission, we use the method used in [Ramsauer et al., 2021] and show that under this setting, the T5 and BERT's attention inference can be seen as solving the proposed convex problem as $\alpha z$ is close to $\lambda^\star$ which corroborates our claim.
>
> **Q4**. Regarding Section 3.1 and 3.2, it does not become really clear what the meaning of z,t,u are n the context of the examples. Very late, the specific formulae are given for BERT and T5, but no interpretation or motivation for this choice is given, other than to match formula (18) ...
>
> **A4**. We would like to highlight that the solution to the convex problem, as presented in Eq (18), is consistent with the original attention mechanisms introduced by Bahdanau et al. (2014) and Luong et al. (2015). These mechanisms encompass most of the essential elements found in the more contemporary transformer architectures. In transformers such as BERT and T5, the implementations offer greater flexibility by making more components trainable. However, this does not alter the fundamental dynamics, as we can demonstrate equivalence through reparameterization. Consequently, we believe that the intuition we provide for deriving Eq (18) also extends to this more general scenario. We have revised the manuscript to clarify our reasoning.
>
> **Q5**. Based on this, it remains unclear to me why the presented viewpoint would be more helpful to understand attention or improve its design compared to previously offered explanations.
>
> **A5**. We have added additional empirical studies for the OT-based attention structure in the revised version. These studies demonstrate that it improves the performance of vision transformers (ViT). We hope this provides a clearer understanding of the usefulness and effectiveness of our framework.

---

> > ### Author Response · Authors · 2024-07-27
> > **Responses (Part 2)**
> >
> > **Q6**. I am not sure what the purpose of Section 5 is and how it is important for the rest of the paper.
> >
> > **A6**. We have enhanced the introductory part of Section 5 to clarify the relationship between this special setting and the subsequent more general case, and to explain how this connection is beneficial.
> >
> >
> > **Q7**. "That is, even if a word is absent from the source sentence, it will still be optimized if it has a similar embedding in T(k)." How can a word be optimized? ..
> >
> > **A7**. We were trying to say, "Even if a word is not in the source sentence, its embedding can still be optimized." We have edited the sentence to avoid the potential confusion.
> >
> >
> > [Ramsauer et al., 2021] Hopfield Networks is All You Need, Hubert Ramsauer and Bernhard Schafl and Johannes Lehner and Philipp Seidl and Michael Widrich and Lukas Gruber and Markus Holzleitner and Thomas Adler and David Kreil and Michael K Kopp and Gunter Klambauer and Johannes Brandstetter and Sepp Hochreiter, ICLR, 2021
> >
> > [Bahdanau et al., 2014] Neural Machine Translation by Jointly Learning to Align and Translate, Dzmitry Bahdanau, Kyunghyun Cho, Yoshua Bengio, ICLR, 2015
> >
> > [Luong el al., 2015] Effective Approaches to Attention-based Neural Machine Translation, Minh-Thang Luong, Hieu Pham, Christopher D. Manning, ACL, 2015

---

> > > ### Comment · Reviewer_P5hp · 2024-08-02
> > > **Thanks for your responses**
> > >
> > > Dear authors,
> > >
> > > thank you for your detailled response and clarifications. I am willing to update my rating, but I have two short questions remaining:
> > >
> > > 1) As you say,  there could be multiple pairs of  $\alpha$ and $z$ that may lead to the same attention architecture. Given that the main contribution of the paper relies on this connection, I wonder if you did abaltion studies that the choice of $\alpha$ and $z$ that you made is the best one to actually match the BERT and T5 models. That is, how would Figure 4 look like for other valid choices? In particualr, for choices where $\alpha$ is large, one would expect that the overlap becomes worse, as your derivation relies on $\alpha$ being small.
> > >
> > > 2) Thank you for the additional experiments with OT-Attention. These are interesting, and improve the paper. Can you add more details on the training an tuning procedures for both models (standard attention, and OT)? Often, improvments can be made only due to more extensive hyperparameter tuning (for example for the learning rate), and thus it is important to give as much details as possible to ensure that the comparison is fair.

---

> > > > ### Author Response · Authors · 2024-08-08
> > > > **Thank you very much for the reply!**
> > > >
> > > > Dear reviewer,
> > > >
> > > > Thank you very much for the reply and for being open to updating the rating. Here are our responses to the remaining questions:
> > > >
> > > > **Q8.** As you say, there could be multiple pairs of $\alpha$ and $\mathbf{z}$  that may lead to the same attention architecture. Given that the main contribution of the paper relies on this connection, I wonder if you did abaltion studies that the choice of $\alpha$ and $\mathbf{z}$ that you made is the best one to actually match the BERT and T5 models. That is, how would Figure 4 look like for other valid choices? In particualr, for choices where $\alpha$ is large, one would expect that the overlap becomes worse, as your derivation relies on $\alpha$ being small.
> > > >
> > > > **A8.** We would like to note that, as mentioned in A3, our goal here is to demonstrate that by selecting an appropriate pair without any intentional adjustments, our framework can effectively explain how attention works. Therefore, we did not tune the value of $\alpha$ to make the results more appealing; instead, we directly set $\alpha = 1$, as we believe this is the most natural selection when no additional information about the model is provided. As noted at the end of Section 6, an arbitrarily small $\alpha$ implies an arbitrarily unreliable $\mu + z$, which should not be the case.
> > > >
> > > > To address the question, when $\alpha$ increases, we do expect the overlap to worsen. This can be seen directly by considering the special case in Section 5, where $\lambda^*$ can be computed in a closed form. Let $c = \alpha z$ (which is fixed for a given NN), and $b = \mu + z$. The discussion in Section 5 shows that $\lambda^ = \frac{1}{\alpha + 1} c$, and thus the ratio $\frac{\\|\lambda^* - \alpha z \\|}{\\|\lambda^*\\|} = \alpha$. Namely, when the preference distribution $u$ is a spherical Gaussian, the ratio is simply $\alpha$. As a result, if we increase $\alpha$, the ratio will increase.
> > > >
> > > > For practical cases like BERT and T5, although $u$ is not Gaussian, this relationship largely holds. Interestingly, even though we set $\alpha = 1$ in our experiments, the ratio is much smaller than one, which somewhat suggests that the model is trained to have a preference distribution that makes the estimation more accurate.”
> > > >
> > > > **Q9.** Can you add more details on the training an tuning procedures for both models (standard attention, and OT)? Often, improvments can be made only due to more extensive hyperparameter tuning (for example for the learning rate), and thus it is important to give as much details as possible to ensure that the comparison is fair.
> > > >
> > > > **A9.** Thank you for your advice! We’ve added more details on tuning the parameters and learning rate in Section 9.
> > > >
> > > > (We apologize for the delayed response; we are currently handling multiple rebuttals simultaneously.)

---

### Decision · Action_Editor_5YCw · 2024-08-15

**Recommendation:** Reject

**Comment:**

This paper studies transformers and the attention mechanism using convex analysis. The results are further validated by language models. While all reviewers saw some novel insights in this submission, they also pointed out some significant issues with the presentation, motivation, and experimental design. These issues cannot be addressed by the authors' revision and require another round of full review.

In particular, the major issues include:
- Presentation: The current structure and storyline cause some confusion to reviewers and most likely other readers as well. Also, some terminology might not be easily understandable for the ML community (e.g. an explanation how the templates relate to key, query, value of attention). This is important because the position of this paper is to help provide explanations to ML researchers and practitioners.
- Experiment: Some important details and ablations are missing. The connection between the theoretical part and the numerical findings needs to be strengthened.

In light of the review comments and my own reading, I recommend rejection and major revision for this submission. I hope the authors will incorporate the review comments into the resubmitted version.

**Audience:**

Of broad interest to transformer-based studies, though the presentation needs to be polished.

**Claims And Evidence:**

Yes. This paper provides theoretical analysis and numerical validations.

**Resubmission Of Major Revision:**

The authors may consider submitting a major revision at a later time.